# Epigenetic alterations affecting hematopoietic regulatory networks as drivers of mixed myeloid/lymphoid leukemia

Leukemias with ambiguous lineage comprise several loosely defined entities, often without a clear mechanistic basis. Here, we extensively profile the epigenome and transcriptome of a subgroup of such leukemias with CpG Island Methylator Phenotype. These leukemias exhibit comparable hybrid myeloid/lymphoid epigenetic landscapes, yet heterogeneous genetic alterations, suggesting they are defined by their shared epigenetic profile rather than common genetic lesions. Gene expression enrichment reveals similarity with early T-cell precursor acute lymphoblastic leukemia and a lymphoid progenitor cell of origin. In line with this, integration of differential DNA methylation and gene expression shows widespread silencing of myeloid transcription factors. Moreover, binding sites for hematopoietic transcription factors, including CEBPA, SPI1 and LEF1, are uniquely inaccessible in these leukemias. Hypermethylation also results in loss of CTCF binding, accompanied by changes in chromatin interactions involving key transcription factors. In conclusion, epigenetic dysregulation, and not genetic lesions, explains the mixed phenotype of this group of leukemias with ambiguous lineage. The data collected here constitute a useful and comprehensive epigenomic reference for subsequent studies of acute myeloid leukemias, T-cell acute lymphoblastic leukemias and mixed-phenotype leukemias.

Research on the pathogenesis of leukemia has traditionally emphasized the role of genetic lesions, but the importance of epigenetic dysregulation is becoming increasingly recognized. Several epigenetic modulators are recurrently mutated in acute myeloid leukemia (AML) and T-cell acute lymphocytic leukemia (T-ALL), including methylation regulators (*DNMT3A*, *TET2*, *IDH1/2*) and histone writers (*EZH2*, *SUZ12*, *KMT2A, KDM6A*)[1,2]. On the other hand, numerous instances of epigenetic dysregulation leading to aberrant expression of proto-oncogenes have been documented, such as the enhancer hijacking leading to *EVI1* overexpression in 3q26-rearranged AML[3,4] or the formation of a super-enhancer driving *TAL1* upregulation in T-ALL[5]. However, recurrent epigenetic events may occur independently of a known genetic lesion, possibly due to the selection of clones that spontaneously acquire these alterations. For example, hypermethylation of *DNMT3A* recapitulates the effects of mutations in this gene[6].

Therefore, genetic characterization of leukemia may be insufficient to identify all critical pathogenic mechanisms. Accordingly, clustering of AML samples by gene expression reveals subgroups that share known genetic lesions, but also others that cannot be linked to any known abnormality[7]. One of such subgroups was later found to be defined by *CEBPA* silencing due to hypermethylation[8]. This *CEBPA*-silenced cluster exhibited a mixed myeloid/T-lymphoid phenotype,

✉ e-mail: b.wouters@erasmusmc.nl; h.delwel@erasmusmc.nl; michael.rehli@klinik.uni-regensburg.de; claudia.gebhard@klinik.uni-regensburg.de

resistance to myeloid growth factors, and possibly poor prognosis. Subsequent analyses unveiled a genome-wide hypermethylation signature that distinguished this subgroup from both AML and T-cell acute lymphocytic leukemia (T-ALL), yet no mutations typically associated with methylation defects[9]. More recently, an AML subtype with similar characteristics and methylation localized to CpG islands (CGIs) was identified, labeled CpG Island Methylator Phenotype (CIMP)[10,11]. We hypothesize that CIMP and *CEBPA*-silenced leukemias are the same entity, hereinafter referred to as CIMP leukemias.

Hypermethylation of CGIs is a frequent event in cancer that often results in silencing of tumor suppressor genes[12]. Although DNA methylation is traditionally associated with transcriptional repression, its cellular functions are, in fact, much more complex[13,14]. Transcriptional repression in the presence of methylation is thought to be caused by (a) impaired TF binding, and (b) recruitment of chromatin remodelers via methyl-binding domain (MBD) proteins[13]. However, many DNA-binding factors have shown the ability to bind methylated sequences, whereas others may be repelled by DNA methylation[15]. A notable example of the latter is CTCF, which plays critical roles as insulator, transcriptional repressor or activator, and architectural protein[16]. Thus, aberrant methylation can disrupt CTCF-dependent boundaries of topologically associating domains (TADs), resulting in dysregulated expression of neighboring genes[17,18].

Leukemias with ambiguous lineage pose substantial challenges for diagnosis and treatment[19]. Mixed phenotype acute leukemias with myeloid and T-lymphoid features (T/M MPAL) are defined as a separate category by the World Health Organization (WHO) and the International Consensus Classification (ICC), based on co-expression of markers such as CD3 and MPO[20,21]. Moreover, a subtype of T-ALL, known as early T-cell precursor leukemia (ETP-ALL), also exhibits a combination of myeloid and lymphoid surface markers[22,23]. Recent studies have shown that ETP-ALL and T/M MPAL are similar at the genetic and epigenetic level[24], suggesting an overlap between these two entities. The emerging question is how CIMP leukemias, originally diagnosed as AMLs, are related to these other categories from a molecular perspective.

Here, we characterize the poorly understood CIMP leukemias by integrating multiple layers of genetic and epigenetic data. This integrative analysis reveals that these are immature leukemias with features from both AML and T-ALL, resembling ETP-ALL, and that hypermethylation results in the silencing of several lineage-specific TFs, accompanied by loss of accessibility at their binding sites. Similarly, methylation impairs CTCF binding, leading to chromatin remodeling and secondary changes in gene expression that contribute to the unique phenotype of these leukemias.

## Results

### Global DNA methylation identifies a distinct group of hypermethylated leukemias

Previous studies in separate AML cohorts independently identified clusters of patients with genome-wide hypermethylation, but no mutations typically related to DNA methylation[8,10]. We profiled the methylome of 16 of these patients together with 49 other primary AMLs and CD34+ cells from 3 healthy donors (Supplementary Data 1). We used methyl-CpG immunoprecipitation coupled with sequencing (MCIP-seq) to assay 71,000 CpG-rich regions, covering 89% of the 28,691 CpG islands in the human genome (Supplementary Fig. 1a). More than half of the MCIP-seq peaks were near transcriptional start sites (TSS) (Supplementary Fig. 1b).

Unsupervised hierarchical clustering (Fig. 1a, Supplementary Fig. 1c) indicated that CIMP leukemia constitutes a separate subgroup with strong hypermethylation, particularly at regions hypomethylated in CD34+ cells. Samples from both studied cohorts (CIMP-EMC, originally *CEBPA*-silenced, and CIMP-UKR) clustered together, supporting the hypothesis that they belong to the same disease entity. This observation was supported by principal component analysis (PCA) and

uniform manifold approximation and projection (UMAP) (Supplementary Fig. 1d). Taken together, these data confirm that CIMP leukemias are a distinct entity characterized by global hypermethylation of CpG-rich regions.

### The epigenetic landscape of CIMP leukemias reveals an intermediate state between T-ALL and AML

To understand the regulatory underpinnings of their lineage ambiguity, we comprehensively profiled the epigenetic and transcriptional landscape of CIMP leukemias, as well as that of T-ALL, AML, and CD34+ cells from healthy donors (Fig. 1b).

Dimensionality reduction of MCIP-seq data showed that CIMP cases exhibit a methylation profile very close to that of most T-ALLs and markedly separate from that of the large majority of AMLs (Fig. 1c, Supplementary Fig. 2a). This was corroborated by Infinium MethylationEPIC array data from a subset of 5 CIMPs analyzed together with publicly available T-ALL[25] and AML[26] data (Supplementary Fig. 2b). In particular, CIMP leukemias clustered close to ETP-ALL, another entity with mixed myeloid/lymphoid features, and to highly methylated T-ALL cases (referred to as CIMP-positive T-ALL). However, CIMP cases in our cohort presented intermediate gene expression between AML and T-ALL, with most displaying stronger similarity to *CEBPA* DM AML than to ETP-ALL (Fig. 1c, Supplementary Fig. 2c). To explain this apparent discrepancy between methylation and transcription, we next analyzed H3K27ac ChIP-seq and ATAC-seq (Fig. 1c, Supplementary Fig. 2c). In both datasets, most CIMPs unequivocally clustered with *CEBPA* DM AML, with only 3 T-ALLs (one of which was ETP) included in the same group. Hierarchical clustering largely confirmed these findings (Supplementary Fig. 2d–g).

In summary, CIMP leukemias are an intermediate entity between AML and T-ALL with a hybrid epigenetic profile in which CpG-poor enhancer chromatin is accessible at myeloid regulatory regions, but transcription is repressed at CGI promoters by DNA methylation.

### CIMP leukemias are genetically heterogeneous

To elucidate whether genetic aberrations lie at the base of CIMP leukemias, we conducted whole exome sequencing (WES). We found frequent single nucleotide variants (SNVs) and indels in *NOTCH1*, *PHF6*, *MED12*, *WT1*, *IKZF1* and *JAK3*, but none were common to all individuals (Supplementary Data 2, Fig. 2a). Recurrent copy number alterations (CNAs) were identified in genes related to leukemia and hematopoiesis, among which deletions of a region containing *NF1*, *EVI2A* and *EVI2B* were particularly frequent (n = 6) (Supplementary Fig. 3c, Supplementary Data 3, 4). Most CIMP patients (9/14) exhibited gains or losses of at least one chromosomal arm (Fig. 2b, Supplementary Fig. 3a, b). RNA-seq data did not reveal any recurrent fusion genes, although a few patients carried fusions previously reported in leukemia[2,27,28] (Supplementary Fig. 3d, Supplementary Data 5). Of note, 6/13 patients analyzed did not harbor any detectable fusion gene.

Altogether, CIMP leukemias constitute a genetically heterogeneous subgroup, defined by epigenetic rather than genetic commonalities. Their mutational profiles are comparable to those of other acute leukemias of ambiguous lineage, especially ETP-ALL (see Supplementary Data 6-8 and Supplementary Results), suggesting there may be some overlap between these entities.

### Transcriptional signatures suggest that the cell of origin is an early progenitor with a possible lymphoid bias

In line with their hybrid epigenetic profile and their previously reported phenotype[8], CIMP leukemias expressed both typical myeloid markers such as CD13, CD33, CD34 and KIT (Fig. 2c) and lymphoid markers like CD7 and CD3 (Fig. 2d). Notably, surface markers used for immunophenotypic classification of ETP-ALL (expression of CD7, absence of CD1A and CD8, weak expression of CD5 and presence of myeloid markers such as CD13, CD33, CD34 and CD177/KIT[23,29]), were

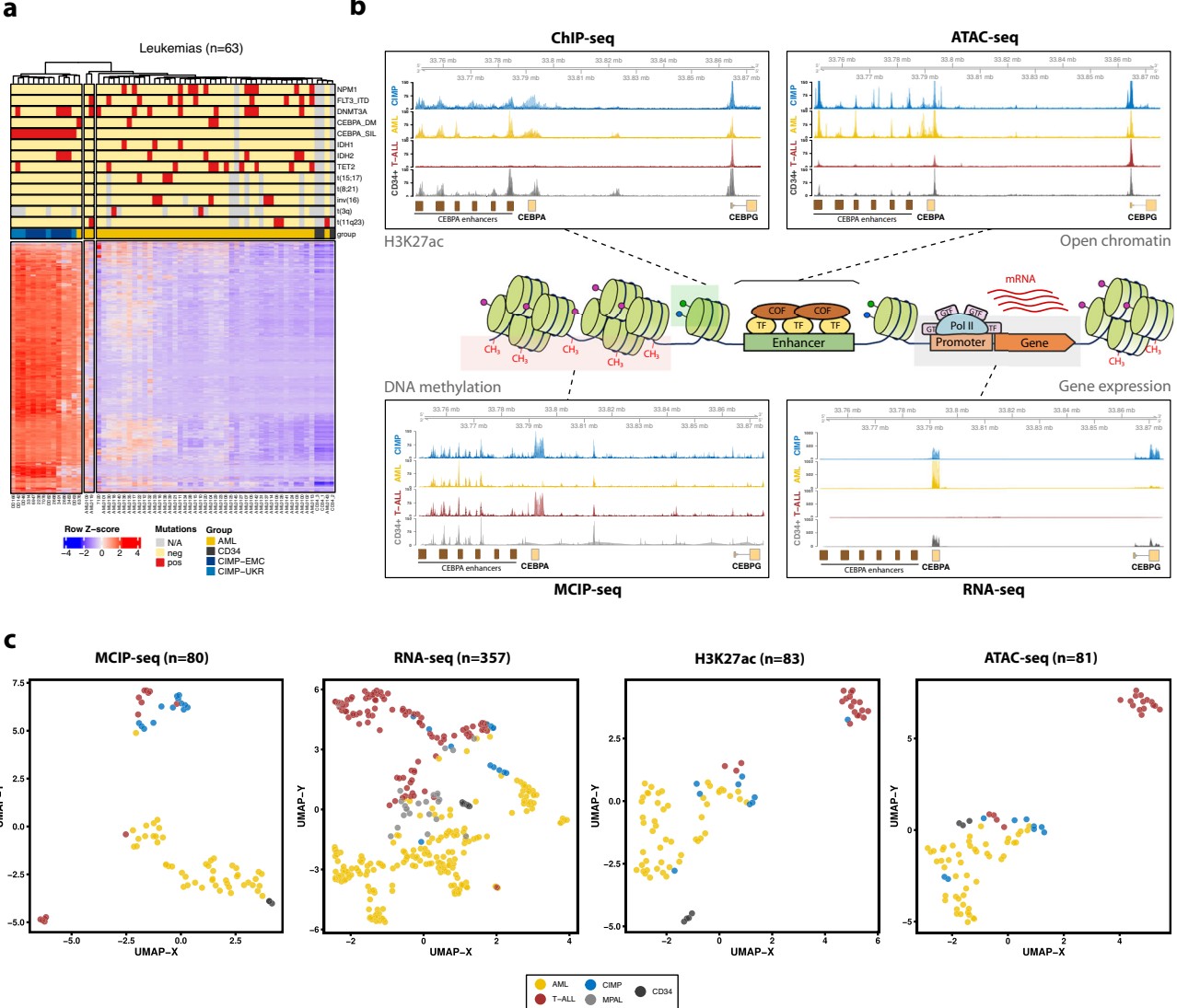

**Fig. 1 | Epigenetic and transcriptional landscape of CIMP leukemias compared to other leukemias and CD34+ cells. a** Heatmap of the 3000 most variable MCIP-seq regions across all samples ($n = 63$), displaying their Z-scores and clustered by Euclidean distance. **b** Diagram depicting different elements of gene regulation and the corresponding data sequenced in this study: MCIP-seq for DNA methylation, H3K27ac ChIP-seq for enhancer activity, ATAC-seq for chromatin accessibility, and RNA-seq for gene expression. **c** Dimensionality reduction with Uniform Manifold

Approximation and Projection (UMAP) of epigenetic and transcriptional data in AML, CIMP, T-ALL and CD34+ HSPCs from healthy donors. From left to right: methylation (MCIP-seq, $n = 80$), gene expression (RNA-seq, $n = 357$), histone H3K27 acetylation (H3K27ac, $n = 83$) measured by ChIP-seq, and open chromatin (ATAC-seq, $n = 81$). Note that individuals did not completely overlap across all datasets; notably, T-ALL patients in the MCIP-seq cohort were not present in any other experiment (see Supplementary Data 1 for details).

expressed accordingly in CIMP, but not those that define T/M MPAL (presence of either MPO or monocytic markers like CD11c, CD14, CD64 or LZE, concomitantly with CD3 expression[20]). To further investigate lineage relationships at the transcriptional level, we conducted gene set enrichment analysis (GSEA) (Fig. 2e, Supplementary Data 9–12). As expected, myeloid gene sets (e.g., *Dick_GMP250*) were downregulated compared to AML, but upregulated compared to T-ALL, the reverse was true for T-lymphoid genes. Interestingly, the top results from the comparison with T-ALL were gene sets derived from ETP-ALL relative to other T-ALLs (e.g., *ETP-ALL_Zhang_Up*, Supplementary Fig. 4a), as well as gene sets related to hematopioietic stem cells (HSCs) (e.g., *Dick_HSC250*). A comparison with CD34+ cells revealed upregulation of T-cell signatures, including ETP, but downregulation of HSC gene sets.

Transcriptional deconvolution with CIBERSORTx[30] showed enrichment for both CLP and GMP signatures, concomitantly with depletion of expression profiles associated with terminally differentiated cells, including T cells, monocytes, and neutrophils (Fig. 2f,

Supplementary Fig. 4b). In a single sample GSEA with a selected number of hematopoietic-related gene sets, the CIMP group exhibited enrichment for HSC genes as well as myeloid and lymphoid signatures halfway between AML and T-ALL (Supplementary Fig. 4c, d).

Taken together, these results suggest that the cell of origin of CIMP leukemias is an early hematopoietic progenitor, possibly committed to the lymphoid lineage. This lymphoid skewing is supported by the upregulation of T-cell signatures relative to HSPCs, but also by the higher similarity with T-ALL in terms of recurrent mutations and methylation profiles. Furthermore, the upregulation of ETP-ALL gene sets further emphasizes the similarity with this other entity of ambiguous lineage.

**Promoter methylation changes correlate with silencing of critical hematopoietic factors in CIMP leukemias**
Next, we investigated the effects of methylation in CIMPs in relation to other leukemias and normal controls using MCIP-seq data. CIMPs

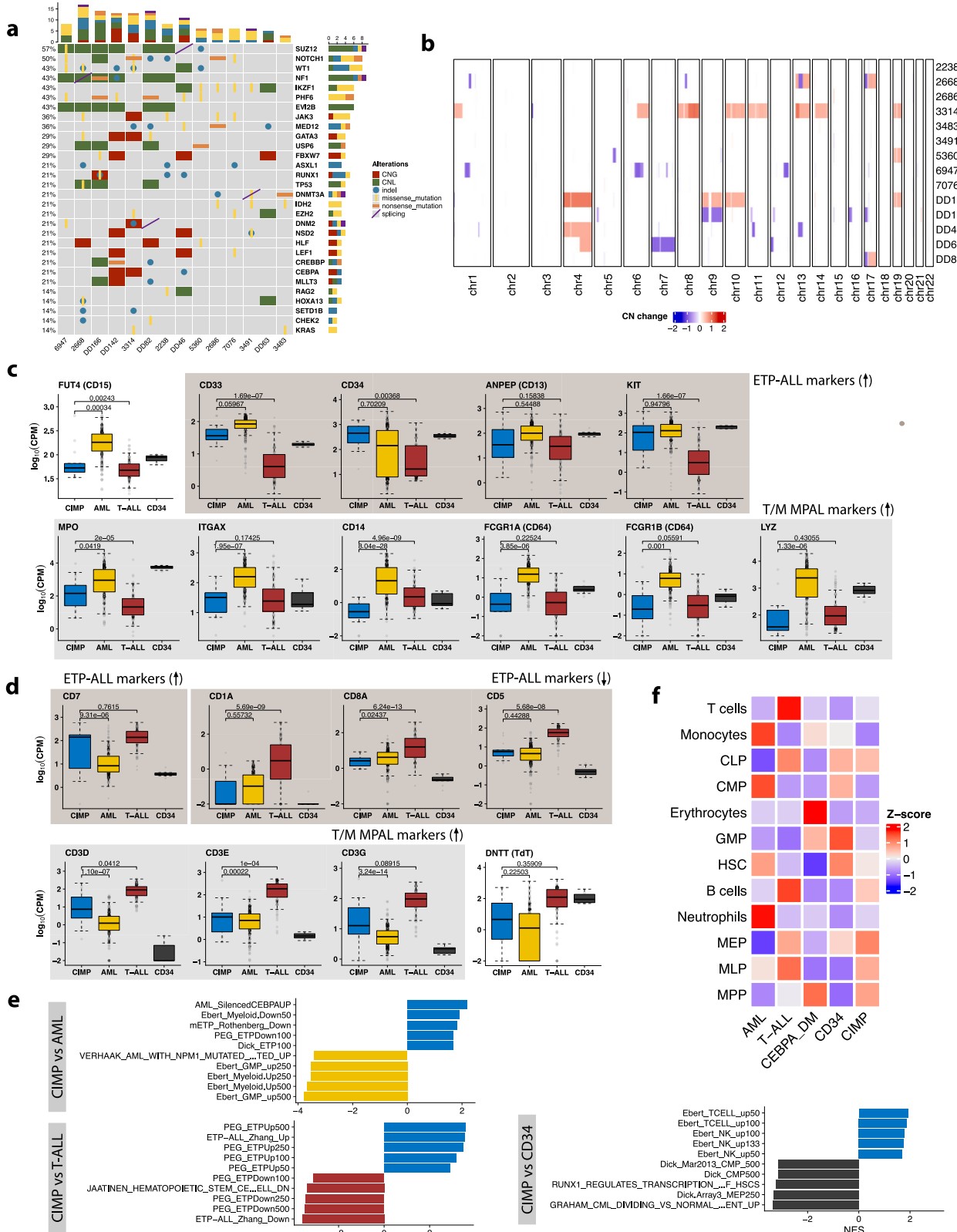

exhibited profound hypermethylation at CGIs, promoters, TSS, and gene bodies compared to AML or HSPCs (Fig. 3a, Supplementary Fig. 5a, b). On the other hand, methylation levels were similar in T-ALL. Promoter methylation levels were the highest in CIMP, followed by T-ALL, AML, and HSPCs (Fig. 3a–c). These differences are consistent with the higher levels of methylation in the lymphoid lineage[31–33]. However, genome-wide hypermethylation was not present

in terminally differentiated cells of those lineages (Supplementary Fig. 5c, d). Differential methylation analysis confirmed extensive hypermethylation in CIMP compared to AML, and to T-ALL to a lesser extent (Fig. 3d, Supplementary Data 13). Methylation array data analogously revealed larger differences between CIMP and AML than T-ALL, particularly at promoters (Supplementary Fig. 5e, f, Supplementary Data 14–16). Interestingly, genes hypomethylated in CIMP relative

**Fig. 2 | The mutational landscape and gene expression signature of CIMP leukemias suggest similarity with ETP-ALL and a very early lymphoid progenitor as the cell of origin. a** Oncoprint displaying single nucleotide variants (SNVs), small inserts and deletions (indels) or copy number alterations (CNAs) affecting genes mutated in at least 2% of the cohort ($n = 14$). Columns correspond to patients and rows correspond to genes, ranked by mutational frequency. Variant calling was performed with an ensemble of tools on whole exome sequencing (WES) data. Different variants are indicated in different colors as shown in the plot legend. CNG = copy number gains; CNL = copy number losses. **b** Heatmap displaying large CNAs in CIMP cases, detected using CNVkit on WES data ($n = 14$). Red indicates a CNG and blue indicates a CNL. **c** Expression of myeloid markers commonly used for leukemia classification in CIMP ($n = 13$), AML ($n = 211$), T-ALL ($n = 100$) and healthy controls ($n = 9$). Markers used to define T/M MPAL or ETP-ALL are indicated. The lower and upper edges of the boxplots represent the first and third quartiles, respectively, the horizontal line inside the box indicates the median. The whiskers extend to the most extreme values within the range comprised between the median and 1.5 times the interquartile range. The circles represent outliers outside this range. The horizontal black lines between boxes represent pairwise comparisons between CIMP

and other leukemias. Statistical significance was determined by a two-sided Wald test in the DESeq2 package and corrected for multiple testing with the Benjamini–Hochberg procedure. **d** Same as **c**, but showing lymphoid markers instead. **e** Bar plot showing the top results from gene set enrichment analysis (GSEA) conducted on a custom version of the MSigDB C2 collection. The analysis was conducted on differentially expressed genes in CIMP relative to AML (top), T-ALL (middle) and CD34+ HSPCs (bottom). **f** Heatmap displaying CIBERSORTx scores for various hematopoietic cell types, using a signature matrix derived from publicly available single-cell RNA-seq[143]. The 25%-trimmed mean of the scores was calculated for each leukemia subgroup (or CD34+ cells), followed by row-wise Z-score normalization. Scores were calculated for every sample and aggregated by disease groups: CIMP ($n = 13$), AML ($n = 189$), CEBPA double mutant (DM) AML ($n = 22$), T-ALL ($n = 100$). The CEBPA DM subgroup was analyzed separately owing to the similarities with CIMP leukemias. CLP common lymphoid progenitor, CMP common myeloid progenitor, GMP granulocyte-monocyte progenitor, HSC hematopoietic stem cell, MEP megakaryocyte-erythrocyte progenitor, MLP multi-lymphoid progenitor, MPP multipotent progenitor.

to T-ALL or ETP-ALL were associated with neutrophil activation, whereas hypermethylated genes were involved in T-cell development (Supplementary Fig. 5h, i, Supplementary Data 17–19). Thus, despite their similarities, CIMP leukemias are slightly more myeloid than ETP-ALLs, which may explain their original diagnosis as AML.

Hypermethylation was more pronounced in regions marked by both H3K4me3 and H3K27me3 (Fig. 3c, Supplementary Fig. 5j, k), typically referred to as bivalent promoters[34]. This is in agreement with reports showing that bivalent promoters are more susceptible to DNA hypermethylation in cancer cell lines and primary tumors[10,35]. GSEA of genes near DMRs confirmed preferential hypermethylation of H3K27me3 targets in CIMP relative to AML, T-ALL, and CD34+ HSPCs (Fig. 3e, Supplementary Fig. 5l, m, Supplementary Data 20–22). This phenomenon may explain the frequent hypermethylation of CGI-associated promoters of genes active in non-hematopoietic lineages, such as neurons (Supplementary Fig. 5g).

Moreover, GSEA detected an enrichment of transcription factor (TFs) genes in methylated regions relative to both AML and CD34+. Besides, some of the DMRs with the strongest increases were adjacent to genes such as *CEBPA*, *LEF1*, *PLK2*, *MEIS1* or *TLE4*, which have a known involvement in either leukemia or hematopoiesis (Fig. 3f, Supplementary Fig. 5n). Integration of gene expression data confirmed that CIMP hypermethylation was accompanied by widespread gene silencing, with methylation levels negatively correlating with gene expression in comparisons with both AML ($r = −0.23$, $p < 2.2 × 10^{-16}$) and T-ALL ($p = −0.18$, $p < 2.2 × 10^{-16}$). A total of 100 TFs with silenced promoters were downregulated relative to AML, including hematopoietic regulators and genes known to be involved in leukemia, such as *CEBPA*, *CEBPD*, *KLF4*, *IRF4* or *MEIS1* (Fig. 4a, b, Supplementary Fig. 6a, b, Supplementary Data 23). Among the few TF genes differentially methylated between CIMP and T-ALL was *LEF1*, which participates in the early stages of thymocyte maturation[36] and is crucial for neutrophilic granulopoiesis[37]. Interestingly, the tyrosine kinase *LYN* was exclusively repressed in T-ALL.

In order to study whether epigenetic changes outside promoters have effects on the transcriptional program of these leukemias, we also integrated RNA-seq with ATAC-seq. We excluded peaks overlapping with a TSS to select putative enhancers, which we then linked to target genes following an ensemble approach. As expected, changes in accessibility at these regions correlated with gene expression in differential analyses between CIMP and AML ($r = 0.17$, $p$ value $< 2.2 × 10^{-16}$) and T-ALL ($r = 0.15$, $p$ value $< 2.2 × 10^{-16}$), albeit only weakly (Fig. 4c, Supplementary Data 24). Similar results were observed for H3K27ac (Supplementary Fig. 6c, Supplementary Data 25). This modest correlation can be explained by multiple factors, such as the multiplicity of enhancers controlling the same promoter, the existence of

concomitant epigenetic processes that participate in gene regulation (especially DNA methylation), and erroneous enhancer-promoter assignment.

In summary, many critical TFs are silenced by methylation in CIMP leukemias, which possibly explains the intermediate transcriptional state of these leukemias, as well as their differentiation arrest.

## Hematopoietic TF networks are rewired in CIMP leukemias

The transcriptional effects of DNA methylation are thought to stem from altered binding affinities for TFs and other regulatory proteins[13]. To elucidate the role of this mechanism in CIMP leukemias, we analyzed the overlap between DMRs and experimentally validated TF binding sites (TFBS) using LOLA on both methylation array and MCIP-seq data. In keeping with the GSEA results, we detected a strong enrichment for binding of the PRC2 complex at hypermethylated regions relative to AML, but also for hematopoietic factors like RUNX1, CBFB, SPI1 or GATA1, as well as CTCF and cohesin (Fig. 5a, Supplementary Fig. 7a left side, Supplementary Fig. 7c, Supplementary Data 26, Supplementary Data 29). Surprisingly, the myeloid TFs CEBPA/B and SPI1 were also overrepresented at hypomethylated positions. When compared to T-ALL, hypermethylated DMRs were strongly enriched for binding sites of lymphoid TFs, notably NOTCH1 and MYB, and transcriptional regulators like RNA Pol II and CTCF (Fig. 5a, Supplementary Fig. 7a, d, Supplementary Data 27 and 30). Together with the fact that hypomethylated DMRs contained SPI1, CEBPB, and CEBPA sites, this suggests a significant number of myeloid regions remain active, thus preserving myeloid potential in CIMP cases. In keeping with this, CIMP leukemias also exhibited increased methylation at NOTCH1, MYB, and TAL1 binding sites than ETP-ALL, and hypomethylation of TFBS for myeloid master regulators such as CEBPA, CEBPB, and SPI1 (Supplementary Fig. 7b, Supplementary Data 28).

To further investigate changes in TF activity linked to lineage commitment, we used *chromVAR* to calculate deviations of chromatin accessibility at TF motifs from ATAC-seq data (Supplementary Fig. 8a, details in Supplementary Results). Chromatin at C/EBP and HLF binding sites were significantly less accessible in CIMP than in AML, whereas RUNX, FOX, and NFATC motifs, among others, were more active (Fig. 5b, Supplementary Fig. 8b, Supplementary Data 31). LEF1, TCF7, and E2F motifs were largely closed in CIMP compared to T-ALL, but those for SPI1 (PU.1) and BACH were more open. In line with these observations, differential accessibility analysis between AML and T-ALL established that accessible C/EBP and SPI1 sites are a hallmark of AML, whereas T-lymphoid leukemias are driven by LEF1, TCF7, and RUNX1, among others. These results were confirmed by footprinting analysis with *TOBIAS*, which predicts TF binding status based on dips in

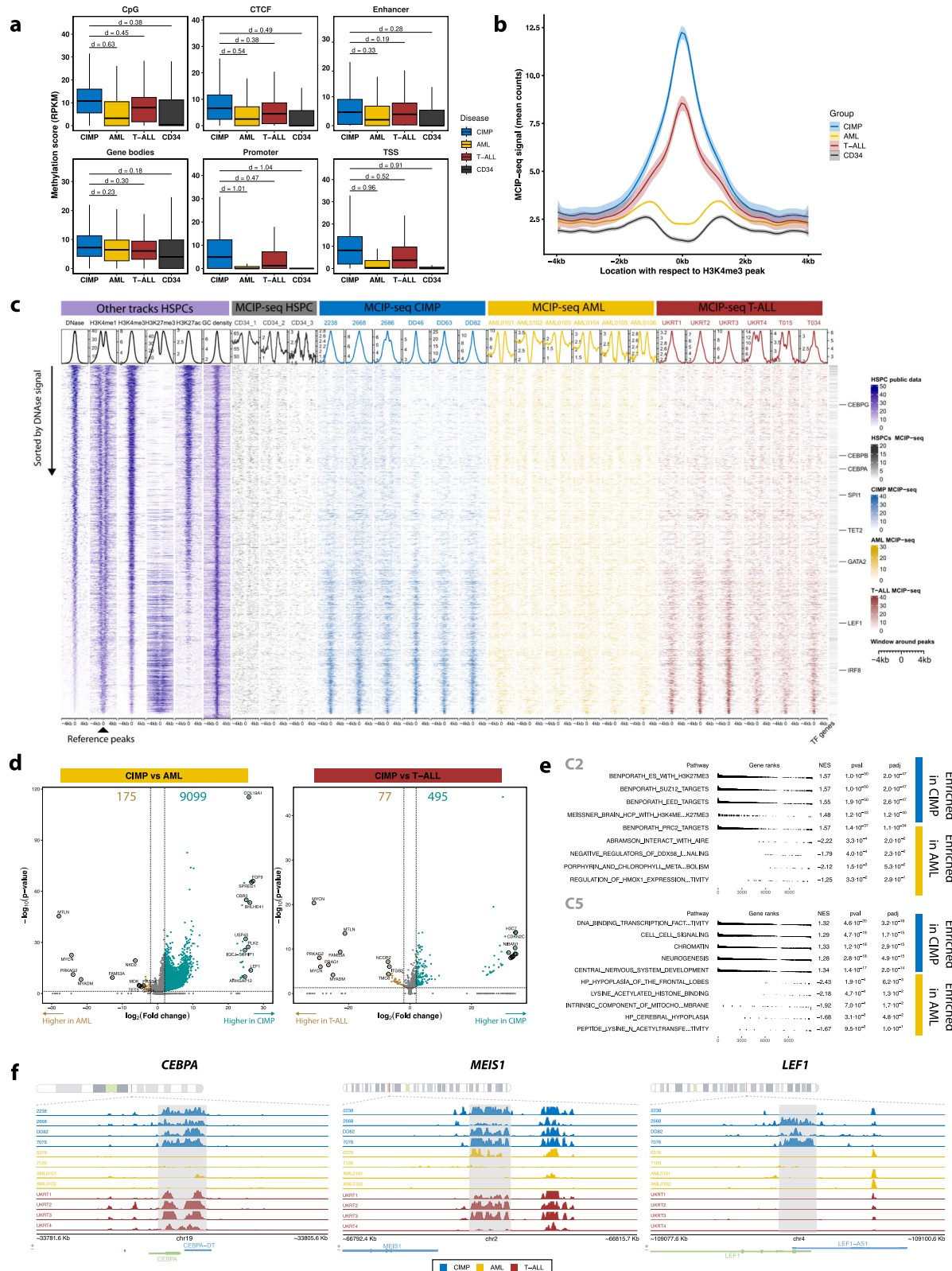

the depth of coverage at their motifs relative to surrounding open chromatin peaks (Supplementary Fig. 8c, d, Supplementary Data 32, 33). In addition to the abovementioned TFs, TOBIAS showed a significant increase in the binding of GATA factors in CIMP compared to AML. Furthermore, this technique also revealed that loss of *KLF4* and *CEBPA* expression is accompanied by reduced binding of TFs at their

promoters, possibly due to hypermethylation (Supplementary Fig. 8e). For validation, we conducted ChIP-seq for CEBPA, SPI1, and TCF7, predicted to be dysregulated. In all three cases, the results were in line with the differential motif activity inferred from open chromatin: CEBPA binding was completely absent in CIMP and T-ALL, and SPI1 exhibited higher binding in CIMP than in T-ALL, whereas the reverse

**Fig. 3 | Functional assessment of methylation differences between CIMP and other leukemias.** All statistical analyses presented here have been performed using MCIP-seq data from CIMP ($n = 13$), AML ($n = 50$), T-ALL ($n = 14$), and healthy CD34+ cells ($n = 3$), unless otherwise specified. **a** Box plot showing methylation levels at different genomic features. The lower and upper edges of the boxplots represent the first and third quartiles, respectively; the horizontal line inside the box indicates the median. The whiskers extend to the most extreme values within the range comprised between the median and 1.5 times the interquartile range. The lines between boxes indicate the effect size as Cohen's $d$, defined as the number of standard deviation units between groups (all comparisons were significant in a two-tailed Welch's $t$ test). Typically, $d$ values below 0.2 are considered small, and above 0.8 are considered large[96]. **b** Average methylation levels of different leukemias and healthy cells at putative gene promoters, defined as 4-kb regions surrounding the center of H3K4me3 ChIP-seq peaks in CD34+ HSPCs. Each line depicts a smoothed average (LOESS function) for a group of patients, with the shaded band indicating the 95% confidence interval. **c** Tornado plots depicting methylation (MCIP-seq) at putative HSPC promoters (H3K4me1 peaks), sorted by chromatin accessibility in

HSPCs (DNase). The color code distinguishes different types of leukemia and HSPCs, and the intensity reflects the degree of methylation. The HSPC tracks in purple were downloaded from ENCODE[78] and show chromatin accessibility (DNase) as well as histone marks for enhancers (H3K4me1), promoters (H3K4me3), activation (H3K27ac) and repression (H3K27me3). GC density was downloaded from the UCSC browser[153]. **d** Volcano plot of differentially methylated regions (DMRs) annotated with the closest genes in the linear genome. The statistical significance of the comparisons between these groups was determined by the Wald test (two-sided) in the DESeq2 package and corrected for multiple testing with the Benjamini–Hochberg procedure. Regions with false discovery rate (FDR) < 0.05 and log2 fold change >2 are highlighted; the numbers at the top indicate the number of differentially expressed genes for each comparison. **e** Summary plot of the top 5 most significant results of pre-ranked GSEA conducted on genes in the vicinity of DMRs between CIMP and AML. The C2 (top) and C5 (bottom) MSigDB collections were used in the analysis. **f** Genomic tracks of MCIP-seq data for a few selected samples of each leukemia (CIMP, T-ALL, AML) at promoters of hematopoietic genes with significant changes in methylation.

---

was true for TCF7 (Supplementary Fig. 9a, b). Of note, ChIP-seq also corroborated that SPI1 binding is indeed lost at the *KLF4* promoter (Supplementary Fig. 9c).

Integration with gene expression across 79 patients where both RNA-seq and ATAC-seq were available revealed that motif activity was largely consistent with gene expression (Fig. 5c). High correlation between expression levels and accessibility identified CEBPB and SPI1 as positive regulators in AML, whereas GATA3 and TCF7 played this role in T-ALL (Fig. 5d, Supplementary Fig. 8f, Supplementary Data 34). Integrated analysis of differential motif accessibility and expression pinpointed a few key TFs that drive the CIMP phenotype: loss of C/EBP proteins and gain of GATA3 separate them from AML, whereas reduced LEF1 and TCF7 prevent lymphoid lineage commitment (Fig. 5e). Interestingly, the motifs for SPI1 and CEBPA were concomitantly active in the same regions and mutually exclusive with LEF1 and GATA3 (Supplementary Fig. 8g).

Despite genome-wide epigenetic and transcriptional alterations (Supplementary Data 9, 13, 35–37), it appears that methylation-mediated silencing of a reduced set of critical hematopoietic TFs is a major driver of the altered phenotype of CIMP leukemias, ultimately impairing their ability to fully commit to either the lymphoid or the myeloid lineage.

## Genome-wide hypermethylation leads to widespread loss of CTCF binding

Since DNA methylation may weaken the binding of CTCF[38,39], the hypermethylation observed at CTCF binding sites (Fig. 3a, Fig. 5a) suggested a possible loss of CTCF binding at those locations. Indeed, CTCF ChIP-seq showed that global CTCF levels were lower in CIMP than in AML and T-ALL (Fig. 6a, b). Differential analysis confirmed widespread loss of CTCF binding in CIMP with respect to AML, and to a lesser extent compared to T-ALL (Fig. 6c, Supplementary Data 38).

For additional insight into the interplay between methylation and CTCF binding, we integrated these data with MCIP-seq, which covered 23% of the CTCF sites detected by ChIP-seq (Supplementary Fig. 11a). As expected, CTCF levels genome-wide were higher in regions with low DNA methylation (Fig. 6b) and gain of DNA methylation in CIMP cases correlated with loss of CTCF binding in the same regions relative to AML (rho = −0.27, $p$ value < $2.2 \times 10^{-16}$) (Fig. 6d, Supplementary Data 39). No meaningful correlation was observed when comparing CIMP to T-ALL, possibly due to insufficient DMRs. CIMP-versus-AML hypermethylation was more pronounced at CTCF binding sites (Supplementary Fig. 11b), especially at those that were lost (Fig. 6e), suggesting those regions are particularly prone to methylation changes.

The invariability of CTCF binding at many other locations (Supplementary Fig. 11c) is in accord with previous studies indicating that only certain CTCF binding sites are sensitive to methylation, such as the ones with CpG in their motif[40,41]. To explore this possibility, we

computed the frequency of CpG dinucleotides at every position of the canonical CTCF motif, which exhibits two peaks at positions 5 and 15, respectively (Supplementary Fig. 11d). CTCF motifs found in regions with loss of CTCF binding and hypermethylation exhibited CpGs at those two positions more frequently than regions where CTCF binding remained unchanged or increased (Supplementary Fig. 11e).

## Loss of CTCF binding is accompanied by changes in 3D organization

Given the prominent role of CTCF in the stabilization of cohesin-mediated chromatin loops[42,43], we conducted in situ Hi-C experiments on CIMP ($n = 9$), AML ($n = 5$), T-ALL ($n = 4$) and HSPCs ($n = 3$) to assess changes in 3D genome organization. Roughly 40–50% of CTCF binding sites overlapped with TAD boundaries and 10-20% with loop anchors, without differences between variable and unchanged peaks (Supplementary Fig. 12a). We detected a clear separation between AML and other leukemias both at the level of TADs (Fig. 7a, Supplementary Fig. 12b) and loops (Fig. 7b, Supplementary Fig. 12c). Most CIMP cases clustered together with T-ALLs, with a few (DD46, DD63) exhibiting stronger similarity with AML. Supervised comparisons of differential loops or interactions (DIs) and variable TADs (ΔTADs) confirmed that differences between CIMP and AML were larger than between CIMP and T-ALL, but smaller than between AML and T-ALL (Fig. 7c, d).

Contrary to our expectations, we did not observe a widespread depletion of chromatin loops or TADs upon loss of CTCF binding in CIMP cases. However, 72% of the loops lost in CIMP relative to AML exhibited decreased CTCF binding in at least one of their anchors, compared to 59% in gained interactions (Fig. 7e). Moreover, the average decrease in CTCF binding was significantly larger in lost interactions (Supplementary Fig. 12e). Therefore, while most changes of chromatin conformation in CIMP seem to occur independently of hypermethylation-derived loss of CTCF binding, the latter has a contributing role. Such effects were not observed in comparisons with T-ALL (Supplementary Fig. 12d, e).

Next, we conducted an unbiased survey of ΔTADs (Supplementary Data 40) and DIs (Supplementary Data 41) with associated changes in CTCF binding and potential implications for gene expression. When comparing CIMP and AML, we found 61 ΔTADs containing differentially expressed genes with loss of CTCF binding at their boundaries and 71 differential enhancer-promoter loops, whose interaction strength strongly correlated with the expression of genes they contacted (rho = 0.67, $p = 1.8 \times 10^{-10}$, Fig. 7f). Among others, loss of insulation was detected at the TADs containing *KLF4* (Fig. 7g) and *CEBPD* (Fig. 7h), both of which also displayed decreased chromatin interactions accompanied by reduced CTCF binding. Interestingly, their promoters were also methylated, suggesting a possible cooperation between distinct epigenetic mechanisms in repression. Examples of

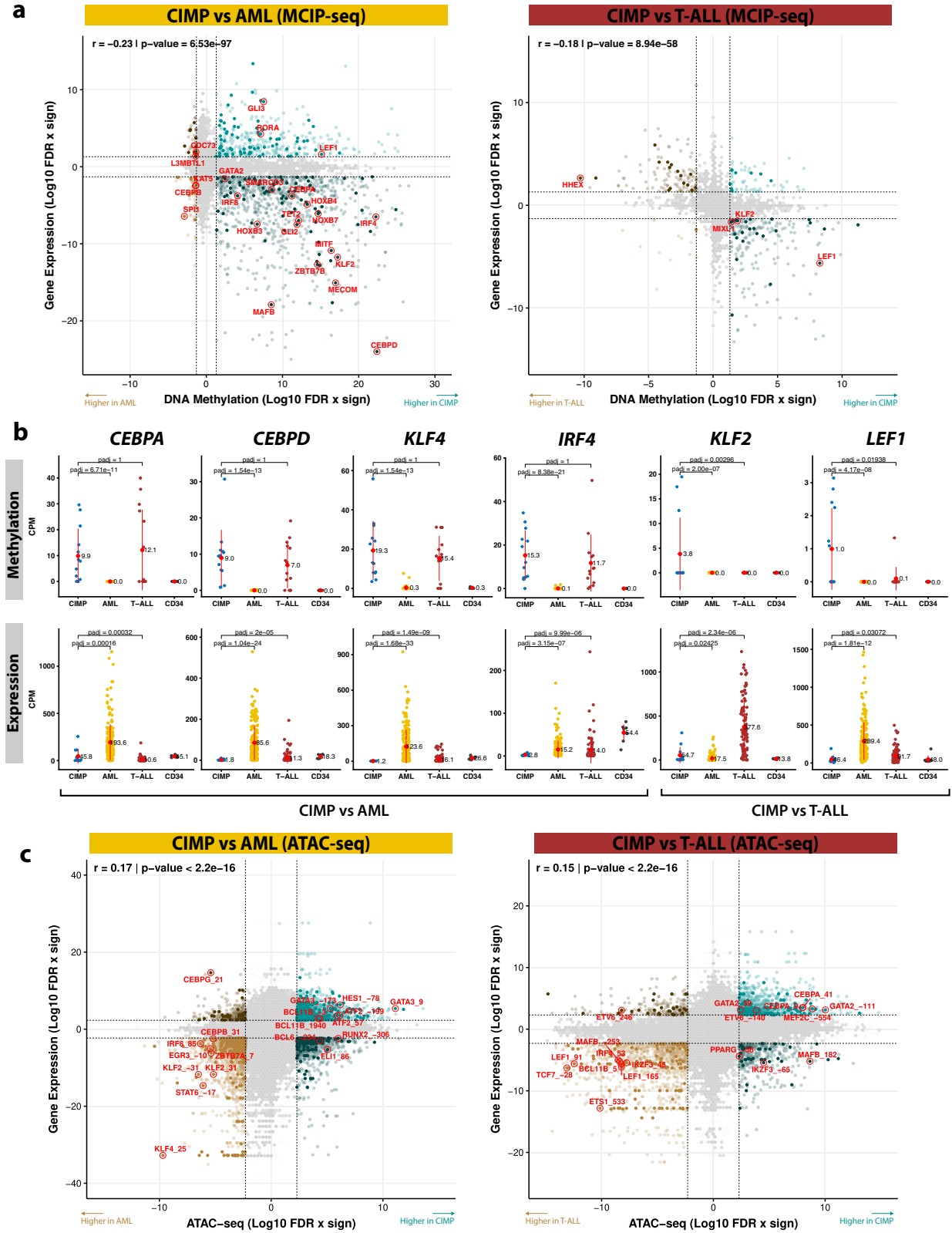

gained enhancer-promoter interactions included a loop connecting *GATA3* with a nearby enhancer element specific to CIMP (Fig. 7i) and a loop involving the promoter of *DNMT3B* (Supplementary Fig. 12i). More details are provided in the Supplementary Results.

In sum, CIMPs exhibit partial rewiring of chromatin interactions when compared to AML, of which only a fraction is attributable to loss of CTCF. However, this mild remodeling correlates with the misexpression of some essential TFs. Very few 3D genome differences could be detected between CIMP and T-ALL, in line with the notion that these leukemias originate from a lymphoid-biased cell.

## Discussion

In this study, we investigated a group of leukemias with a CpG Island Methylator Phenotype (CIMP) and mixed myeloid/lymphoid features.

**Fig. 4 | Integration of methylation and gene expression data reveals silencing of hematopoietic-related transcription factors. a** Starbust plot depicting changes in gene expression (RNA-seq, *Y* axis) and methylation (MCIP-seq, *X* axis) between CIMP and AML (left) and T-ALL (right). The statistical significance of the comparisons between these groups was determined by the Wald test (two-sided) in the DESeq2 package and corrected for multiple testing with the Benjamini–Hochberg procedure. The values are the log10 of the false discovery rate (FDR) with the sign of the fold change. Genes with FDR < 0.05 and log2 fold change >2 for both data types are colored (turquoise if hypermethylated, brown if hypomethylated). Those with non-significant changes are binned into gray hexagons whose opacity is proportional to the number of genes therein. Genes encoding for transcription factors are shown in solid color, among which those involved in hematopoiesis are highlighted in red (GO term: 0030097) and labeled. The rest of the genes are semitransparent. The Pearson correlation coefficient (*r*) and its related *p* value (two-sided) for the relationship between methylation and expression are shown at the top left. Sample sizes for MCIP-seq and RNA-seq were, respectively: 13/13 (CIMP), 50/211 (AML), 14/100 (T-ALL). **b** Jitter plots showing methylation (top) and expression (bottom) of a few selected genes in CIMP, other leukemias, and HSPCs. The central red dot indicates the mean and the vertical red lines correspond to the standard deviation. The horizontal black lines represent pairwise comparisons between CIMP and other leukemias, obtained with the same statistical methodology and sample sizes described in **a**. **c** Starbust plot depicting changes in gene expression (RNA-seq, *Y* axis) and chromatin accessibility at enhancers (ATAC-seq, *X* axis) between CIMP and AML (left) and T-ALL (right). The statistical methodology and the color code are the same as in **a**, but for chromatin accessibility instead of DNA methylation. Note that a single gene may be targeted by multiple enhancers, each of which is labeled based on the distance with respect to the TSS. Sample sizes for ATAC-seq and RNA-seq data were, respectively: 9/13 (CIMP), 51/211 (AML), and 19/100 (T-ALL).

We established that this phenotype is the consequence of a hybrid epigenetic landscape that integrates methylation patterns similar to those of T-ALL with a regulatory landscape that retains significant myeloid potential, resulting in an intermediate transcriptional program. In the absence of common genetic lesions, this shared epigenetic profile seems to be the main defining feature of CIMP leukemias.

The existence of acute leukemias with a mixed myeloid/lymphoid phenotype has long been recognized[44]. The 2022 WHO classification and the ICC identify T/M MPALs based on a reduced number of immunophenotypic markers[20,21], but the CIMP cases identified here do not always conform to these criteria (Fig. 2c, d). Moreover, the mutational profile of MPALs does not exactly match our findings[20]. In the end, T/M MPAL is a broad category that may partially overlap with CIMP, but possibly encompasses multiple subtypes of leukemia with variable cells of origin and pathogenic mechanisms. Another well-known leukemia with ambiguous lineage is ETP-ALL[22], defined initially by a gene expression signature derived from murine ETPs, but typically identified by associated membrane markers. Although CIMP and ETP-ALL cluster separately in genome-wide analyses of epigenetic and transcriptional data, they share immunophenotypic markers and mutational profiles. Therefore, CIMP and ETP-ALL may be related disease entities in a spectrum ranging from myeloid to lymphoid phenotypes, ultimately distinguished by their epigenetic features.

The hybrid epigenetic state of CIMP leukemias and their mixed phenotype invites the question of what their cell of origin is. DNA methylation is a stable mark of epigenetic memory that maintains cell identity across cell divisions[45], which has been exploited to predict cell types[32] and identify the cell of origin in various cancers[46,47]. The hypermethylation in CIMP and many T-ALLs with respect to AML could relate to the higher baseline activity of DNA methyl transferases in the lymphoid lineage[31–33]. On the other hand, open chromatin and H3K27ac indicate proximity to the myeloid lineage, whereas transcriptional data paints an intermediate picture. The inconsistency between different analyses is a likely consequence of phenotypic plasticity and the heterogeneity of these leukemias, some of which appear more myeloid (e.g., #DD166, #3491). The emerging conclusion from these results and the transcriptional signatures detected by CIBERSORTx and GSEA is that CIMPs are likely to stem from an early progenitor, possibly lymphoid-primed, but with the capability to differentiate into myelo-erythroid cell types. Although the specific cell of origin remains uncertain, the lymphoid-primed multipotent progenitor (LMPP) compartment is compatible with these observations. Of note, Zhang et al. reported that ETP-ALLs are enriched for GMP and HSC gene sets and thus derive from HSPCs, rather than ETPs as initially thought[24]. This is congruent with our observations in CIMP, once again underscoring the similarity between these entities, and suggests they both derive from very early lymphoid progenitors.

Aberrant methylation results in silencing of several critical TFs involved in myeloid lineage specification, including *CEBPA*, *KLF4* or

*IRF4*. Interestingly, *IRF4* and a few genes like *MAFB* (another inducer of monocytic maturation) or *KLF4* are completely repressed in CIMP, whereas they remain active in some T-ALL cases. Some TFs involved in lymphopoiesis are also exclusively silenced in CIMP leukemias, such as *LEF1*, a nuclear mediator of WNT signaling that regulates early stages of thymocyte maturation[36] and represses CD4 + T-cell programs in CD8 + T cells[48]. Deletion of *LEF1* results in the upregulation of non-T-lymphoid genes via genome reorganization[49], which could contribute to the mixed phenotype observed here. Accompanying these transcriptional changes were alterations in methylation and chromatin accessibility at binding sites of critical myeloid (CEBP family, SPI1), lymphoid (NOTCH1, LEF1, TCF7) and other hematopoietic (BACH2, HLF, IRF4) factors. Incidentally, hypermethylation of SPI1 binding sites has been reported as a driver of leukemogenesis in TET2-mutated AML[50]. Altogether, hypermethylation of key TFs and their binding sites in CIMP leukemias dysregulates hematopoietic networks, suspending differentiation at an intermediate, unresolved epigenetic state.

Several lines of evidence suggest that the loss of C/EBP TFs is a cornerstone of the leukemogenic process in CIMP leukemias. Firstly, all members of the family, except for the antagonistic *CEBPG*, are silenced by promoter hypermethylation. Secondly, chromatin accessibility at regions containing C/EBP motifs is drastically reduced. Thirdly, CIMP exhibits epigenetic similarity with AML subtypes in which *CEBPA* is either repressed or dysfunctional, namely t(8;21) AML and *CEBPA* DM AML. Given that *CEBPA* promotes myelopoiesis at the expense of lymphoid commitment[51], its inactivation without compensation is likely to be a common driver of differentiation block in both CIMP and AML. Interestingly, the +42-kb enhancer that drives *CEBPA* expression in myeloid cells[52] is active in both CIMP and AML, but absent in T-ALL (Supplementary Fig. 13a, b). It is thus tempting to speculate that transformation took place in a cell type that would normally be primed to express *CEBPA*, once again pointing to an early progenitor that is only biased towards the lymphoid lineage, but retaining multilineage priming.

DNA hypermethylation was also pronounced at CTCF binding sites, which was accompanied by widespread loss of CTCF binding. Since many of these sites co-located with loop anchors and TAD boundaries where CTCF stabilizes cohesin-mediated interactions, we expected a major impact on 3D genome organization. However, this was not the case. A possible explanation is that CTCF loss does not necessarily abolish TADs. While total depletion of CTCF does lead to a global loss of TADs[53], alteration of a single CTCF site may[54,55] or may not[56,57] be sufficient to perturb a TAD boundary. This is partially because many TAD boundaries harbor clusters of redundant CTCF binding sites that confer them resilience to small changes[58,59], but also to the existence of alternative mechanisms that preserve TAD boundaries[57]. Depletion of CTCF must be near complete for a significant impact on TAD insulation[53], which explains why the limited loss due to methylation changes results in mostly modest changes. On

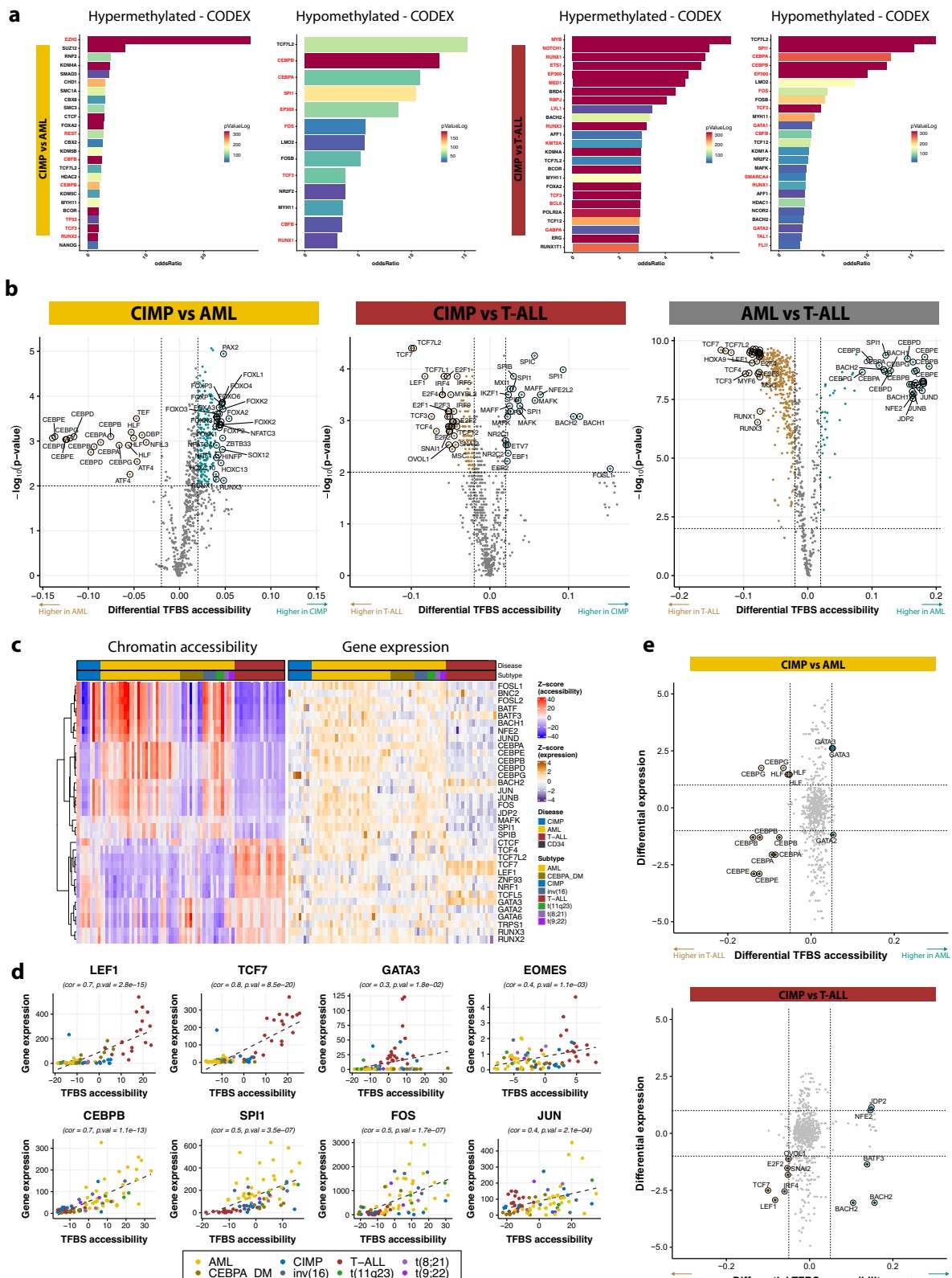

the other hand, although CTCF is present at the vast majority of TAD boundaries, it is only found at a small fraction of enhancer-promoter loops[42], which are frequently occupied instead by YY1[60,61]. Besides, the reduced number of differential interactions identified may be a consequence of the limited sample size and resolution of this Hi-C dataset.

Nonetheless, hypermethylation-driven CTCF loss modulates 3D organization at specific loci, in keeping with previous studies[17,18]. This phenomenon may be complemented by changes in TF genes like *LEF1*, which also modulates chromatin interactions[49]. A striking example is the disruption of several loops and TAD insulation at the *KLF4* locus, presumably abolishing the interaction between its promoter and putative enhancers, which is inactive in CIMP. The lost CTCF binding site that normally stabilizes these loops is at the *KLF4* promoter, which is hypermethylated. Among its multiple roles in hematopoiesis[62], KLF4

**Fig. 5 | Disturbance of hematopoietic regulatory networks in CIMP leukemias suspends both hematopoiesis and lymphopoiesis. a** Bar plot depicting enrichment for experimentally confirmed TF binding sites from the CODEX database at differentially methylated regions between CIMP and AML (left) or between CIMP and T-ALL (right), as derived from EPIC array data (CIMP: $n = 5$, AML: $n = 272$, T-ALL: $n = 119$). Enrichment was calculated with a one-sided Fisher´s exact test using the *LOLA* R package. The length of the bars corresponds to the odds ratio and their color to the $-\log10(p$ value); only a maximum of 25 results with a $-\log(p$ value) above 50 are shown. TFs involved in hematopoiesis hematopoiesis (GO term: 0030097) are highlighted in red. **b** Volcano plots displaying comparisons in motif activity between CIMP and AML (left), CIMP and T-ALL (middle) or AML and T-ALL (right), as derived from ATAC-seq data (CIMP: $n = 9$, AML: $n = 51$, T-ALL: $n = 19$). Motifs with a *p* value < 0.01 (Wilcoxon signed-rank test, two-sided) and |differential deviation|> 0.01 are colored. **c** Heatmap depicting TFBS accessibility and TPM-normalized gene expression in those patients where both RNA-seq and ATAC-seq

were available ($n = 79$). Only the top 50 most variable motifs were selected, aggregated as 35 TFs after excluding alternative motifs for the same protein. **d** Correlation between motif activity as estimated by chromVAR and TPM-normalized gene expression in every individual where both RNA-seq and ATAC-seq data were available ($n = 79$). Sample sizes for each leukemia are described in **b**. The Pearson correlation coefficient ($r$) and its related *p* value (two-sided) for the relationship between TFBS accessibility and gene expression are shown at the top. A positive correlation suggests that the TF is a bona fide driver of chromatin accessibility at its predicted binding sites; a negative correlation may indicate either repression or a competitive effect. The scatter plots in this figure show some of the most relevant TFs with positive correlation in myeloid and lymphoid leukemias. The full dataset is available at Supplementary Data 34. **e** Starbust plot depicting changes in TFBS accessibility inferred by *chromVar* (X axis) and gene expression (Y axis) between CIMP and AML (left) and T-ALL (right). Sample sizes for CIMP, AML, and T-ALL were 9, 51, and 19, respectively.

is required for monocyte differentiation[63], whereas its downregulation is required for lineage commitment of T cells[64]. During these processes, KLF4 stimulates the formation of open chromatin and directly establishes de novo chromatin loops independently of CTCF[65,66], possibly explaining changes in the 3D structure of CIMP leukemias that do not co-occur with variations in CTCF binding. Inactivation of *KLF4* by promoter methylation has been previously reported in T-ALL[67] and chronic lymphocytic leukemia (CLL)[68], as inhibition of T-cell genes by KLF4 impairs T-ALL progression[69]. Thus, the complete loss of *KLF4* in CIMPs potentially contributes to a blockade of the myeloid trajectory while enabling the expression of lymphoid genes. Notably, the expression of KLF4 in CLL can be rescued by inhibition of NOTCH1, which is frequently mutated in CLL[68]. As 43% of the CIMP cases also exhibit such activating mutations, targeting NOTCH1 can be an attractive therapeutic avenue for these leukemias.

The mechanisms underlying aberrant methylation in CIMPs are uncertain. None of the recurrently mutated genes in this leukemia have any known involvement in the methylation machinery. However, the expression of *TET2* was significantly downregulated relative to AML due to promoter hypermethylation, whereas *DNMT1*, *DNMT3A*, and *DNMT3B* were slightly upregulated by either demethylation or gained chromatin interactions. That is, aberrant methylation could result from the inactivation of demethylating enzymes coupled with an increase in de novo and maintenance methylation. As mentioned above, another likely possibility is that the methylation signature of CIMP leukemias is partially inherited from their cell of origin, explaining the similarity with a subset of T-ALLs. A distinctive feature of this aberrant methylation is that it preferentially localizes to bivalent promoters, in keeping with reports that bivalent promoters are susceptible to DNA hypermethylation in cancer[35]. One possible explanation is that H3K4me3, which protects bivalent promoters against DNA methylation by DNMT3A[70,71], is lost in these regions (Supplementary Fig. 13d). Moreover, DNMT3A has been reported to associate with PRC2, which could lead to hypermethylation of H3K27me3-marked domains in the absence of protective H3K4me3[72]. This interaction could be facilitated by the lack of expression of DNMT3L (Supplementary Data 9), which competes with DNMT3A and DNMT3B for interaction with PRC2[73].

In conclusion, CIMP leukemias are a group of immature leukemias of ambiguous lineage whose mixed phenotype reflects a hybrid epigenomic landscape, with methylation patterns of lymphoid leukemias superimposed on an enhancer repertoire that preserves a large degree of myeloid potential (Supplementary Fig. 13e). The repression of *CEBPA* likely plays a key role in locking out the myeloid lineage, while the formation of new loops enables the expression of T-cell genes like *GATA3*. At the same time, silencing of other TFs required for T-cell commitment, such as KLF4, prevent terminal differentiation of T cells. Taken together, this study provides a detailed picture of the unique epigenomic landscape of CIMP leukemias and identifies potential mechanisms driving their differentiation arrest. Furthermore, the data

collected here constitute a useful epigenomic reference for subsequent studies in AML, T-ALL, and mixed phenotype leukemias.

## Methods

### Ethical statement

Our research complies with all relevant ethical regulations and was approved by the Medical Ethical Committee of the Erasmus University Medical Center (Medisch Ethische Toetsings Commissie). All patients provided written informed consent in accordance with the Declaration of Helsinki.

### Patient material and data generation

Samples of AML, CIMP, T-ALL patients and healthy donors were collected from the biobanks of the Erasmus MC Hematology department (Rotterdam, The Netherlands), the University Hospital Regensburg Internal Medicine department (Regensburg, Germany) and the University Hospital Carl Gustav Carus (Dresden, Germany). Mononuclear cells were isolated from bone marrow or peripheral blood as described previously[7]. A summary of the data generated for each patient or donor is available in Supplementary Data 1.

### Data generation and processing

**Methyl-CpG immunoprecipitation sequencing (MCIp-seq).** To measure methylation, we employed Methyl-CpG-immunoprecipitation (MCIP) a technique that relies on a fusion protein consisting of the methyl-binding domain (MBD) of MBD2 and the Fc portion of IgG1 to detect methylated regions, exploiting the natural preference of MBD for 5-methylcytosine (5-mC)[74]. MCIP-seq was performed using the EpiMark® Methylated DNA Enrichment Kit (NEB, Frankfurt, Germany) according to the manufacturer´s guidelines. In brief, genomic DNA was fragmented to an average size of 200 bp using the sonication system Covaris S220 (Covaris, Woburn, USA). Each sample (200 ng) was incubated with 15 µl MBD2-Fc/Protein A magnetic beads and incubated for 1 h at room temperature. Unbound DNA was washed off with washing buffer containing 500 mmol/L NaCl. Captured methylated DNA was recovered by adding 50 µl DNAse free water and incubation at 65 °C for 15 minutes. The distribution of CpG methylation densities in both fractions (unmethylated and methylated) was controlled by qPCR using primers covering the imprinted SNRPN and a genomic region lacking CpGs (empty 6.2). Sequencing libraries were prepared with the NEBNext Ultra II DNA Library Prep Kit for Illumina (NEB) according to the manufacturer's instructions. The quality of dsDNA libraries was analyzed using the High Sensitivity D1000 ScreenTape Kit (Agilent) and concentrations were determined with the Qubit dsDNA HS Kit (Thermo Fisher Scientific). Libraries were single-end sequenced on a HiSeq 3000 (Illumina).

MCIP-seq reads were aligned to the human reference genome build hg19 with *bowtie*[75] (v1.1.1) and bigwig files were generated for visualization with *deepTools bamCoverage*[76] (v3.5.1) and the options

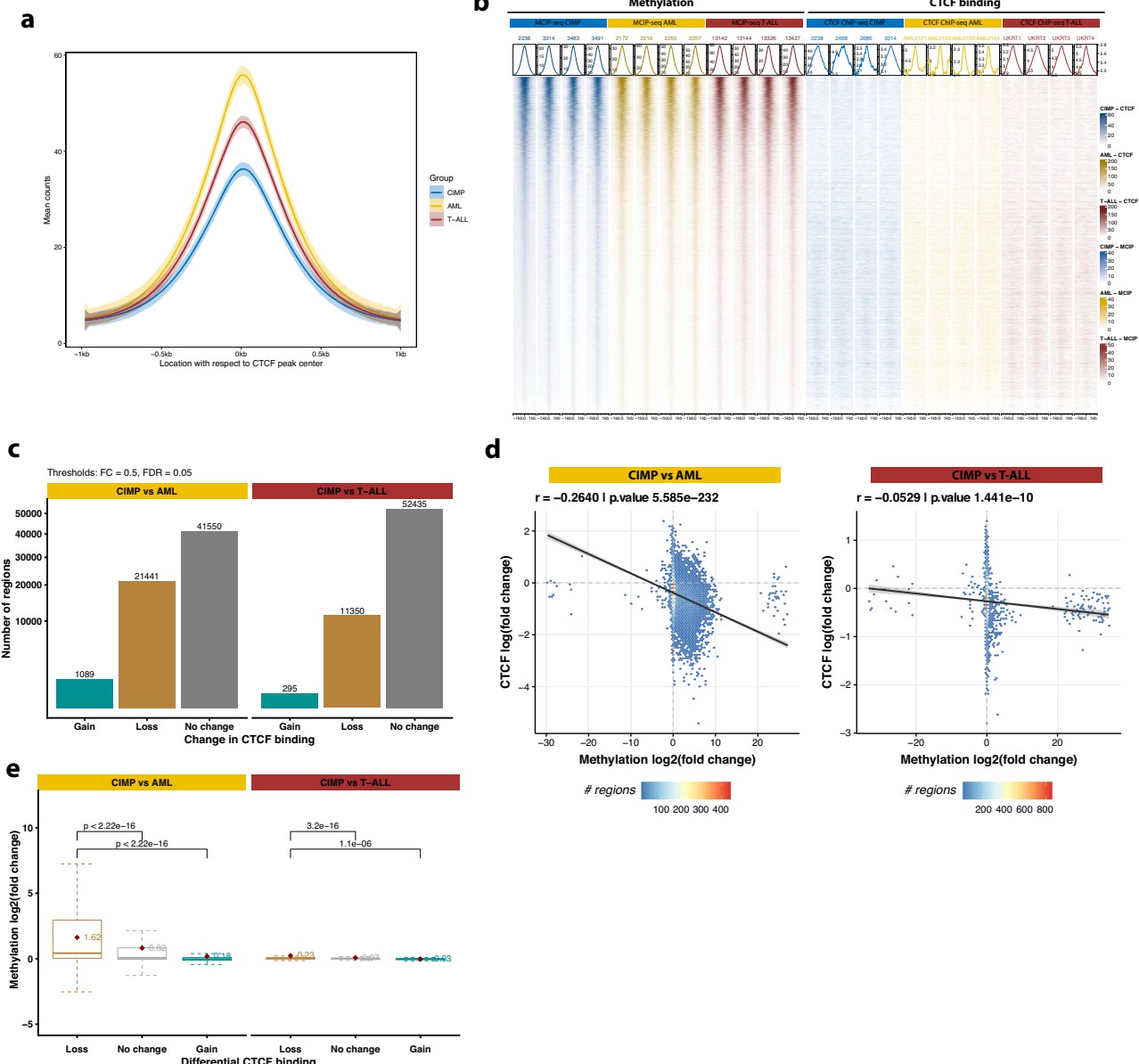

**Fig. 6 | Hypermethylation in CIMP leukemias leads to loss of CTCF binding.**
Statistical analyses of CTCF ChIP-seq data shown here were conducted with the following sample sizes ($n$): CIMP = 9, AML = 10, T-ALL = 19. **a** Average CTCF binding in 1-kb regions surrounding the center of CTCF ChIP-seq peaks on a consensus master list. Each line depicts a smoothed average (LOESS function) for a group of patients, with the shaded band indicating the 95% confidence interval. **b** Tornado plots depicting methylation levels and CTCF binding at the 25,000 most variable CTCF peaks found in at least 4 patients of the entire cohort. Four representative samples of each leukemia type (CIMP, AML, and T-ALL) are presented. The plot above shows the average signal around the center of the peaks for each patient. An inverse correlation between methylation and CTCF binding can be observed. **c** Bar plot of differentially methylated regions (DMR) in supervised comparisons of MCIP-seq peaks between CIMP and AML (left) or T-ALL (right). A threshold of FDR < 0.05 and |log2 FC|>1 was used to determine significant DMRs. NS not significant. **d** Hexagonal heatmap showing the inverse correlation between differences in promoter methylation ($X$ axis) and differences in CTCF binding ($Y$ axis) in CIMP

compared to AML (left panel) and T-ALL (right panel). Data are binned, with the color scale indicating how many promoters are contained in each bin. The values correspond to the log2 of the fold change between conditions, calculated by DESeq2. A regression line is depicted in black, with a shaded gray band indicating the 95% confidence interval. The Pearson correlation coefficient ($r$) and its related $p$ value (two-sided) for the relationship between DNA methylation and CTCF binding are shown at the top left. **e** Box plot displaying methylation changes in relation to differences in CTCF binding between CIMP and AML (left) or T-ALL (right). The lower and upper edges of the boxplots represent the first and third quartiles, respectively; the horizontal line inside the box indicates the median. The whiskers extend to the most extreme values within the range comprised between the median and 1.5 times the interquartile range. The horizontal black lines on top represent pairwise comparisons between groups, with a $p$ value derived from a two-sided Wilcoxon test. No multiple correction adjustment was used. The number of CTCF peaks in each category (loss, no change, or gain) is shown in **c**.

--normalizeUsing RPKM --smoothLength 100 --binSize 20. Peak calling was performed with MACS2[77] (v2.2.7.1) using default settings and input DNA as a control. The resulting peaks were filtered against the ENCODE blacklisted regions[78]. Furthermore, a list of regions accessible by MCIP-seq was defined based on data from monocytes

treated with the CpG Methyltransferase SssI. All peaks not overlapping with this list of mappable regions were considered false positives and discarded using *bedtools intersect*[79]. Functional annotation of peaks was performed with the *ChIPseeker* (Supplementary Fig. 1a) and the *annotatr*[80] (Supplementary Fig. 1b) R packages. The

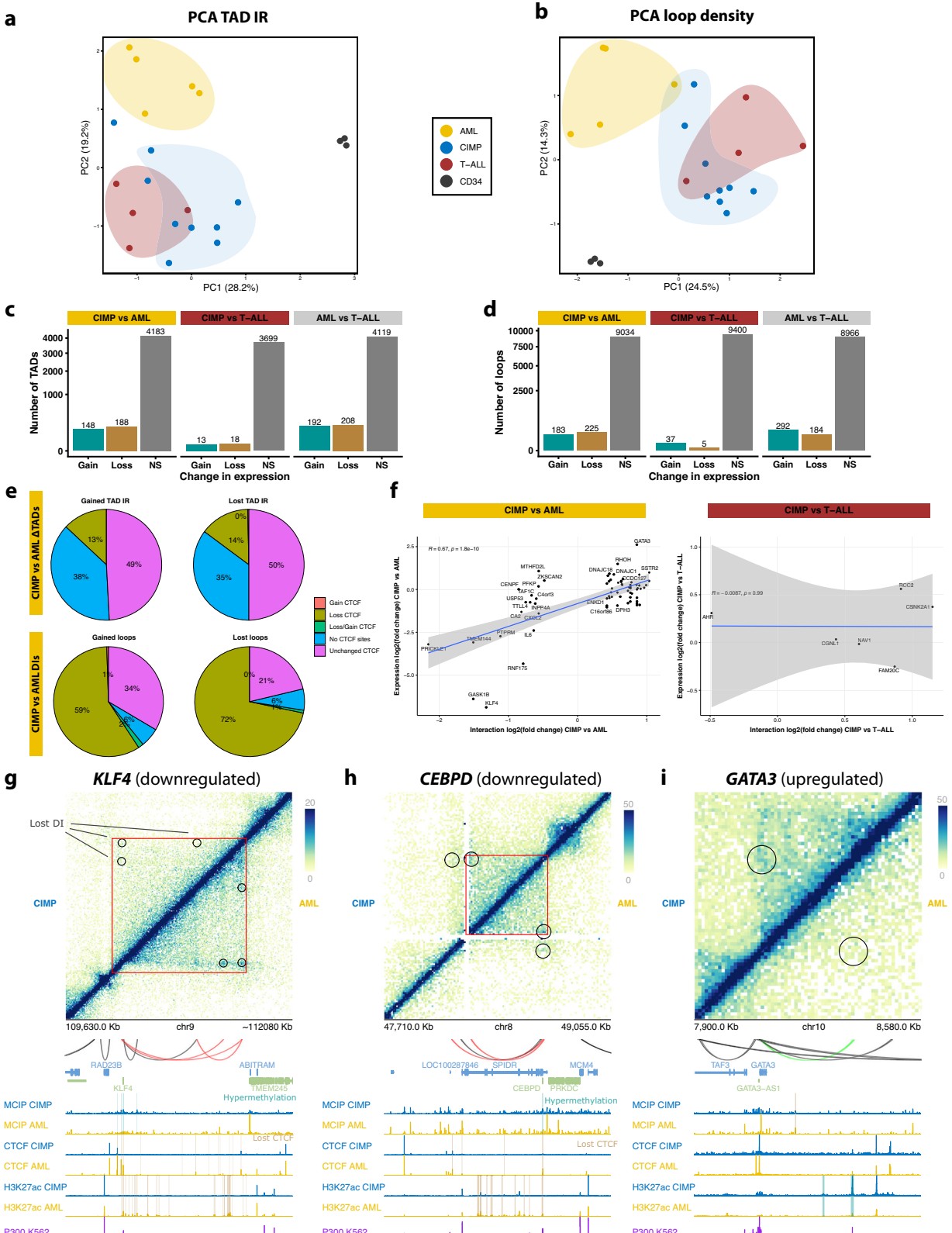

assay detected 71,000 CpG-rich regions with an average length of 650 base pairs (bp), covering 89% of the 28,691 CpG islands in the human genome (Supplementary Fig. 1a).

**RNA sequencing (RNA-seq).** DNA and RNA were isolated using the AllPrep DNA/RNA mini kit (Qiagen, #80204). RNA was converted into cDNA using the SuperScript II Reverse Transcriptase (Thermo Fischer

Scientific) according to standard diagnostic procedures. Sample libraries were prepped using 500 ng of input RNA according to the KAPA RNA HyperPrep Kit with RiboErase (HMR) (Roche) using Unique Dual Index adapters (Integrated DNA Technologies, Inc.). Amplified sample libraries were paired-end sequenced (2 × 100 bp) on the Novaseq 6000 platform (Illumina) and aligned against the human genome (hg19) using *STAR* v2.5.4b[81].

**Fig. 7 | Chromatin interaction landscape of CIMP and other leukemias.** Differential analyses of Hi-C data in **d**–**g** were performed with DESeq2 between CIMP (*n* = 8) and either AML (*n* = 5) or T-ALL (*n* = 4). **a** PCA plot of TAD inclusion ratios (IR) calculated by HOMER in Hi-C data from CIMP (*n* = 8), AML (*n* = 5), T-ALL (*n* = 4) and CD34+ cells (*n* = 3). **b** PCA plot of loop density scores calculated by HOMER in Hi-C data. **c** Bar plot showing TADs with differential IRs between different leukemias as calculated by DESeq2. Only TADs with a log2 fold change larger than 0 and FDR < 0.05 have been considered. **d** Same as **c**, but for loop density scores. **e** Distribution of gains or losses in CTCF binding in variable TADs (top) or differential interaction (bottom) when comparing CIMP vs AML. Lost DIs are enriched for sites with decreased CTCF binding. **f** Correlation between changes in interaction strength and expression levels of the genes contacted by those loops in CIMP vs AML (left) and CIMP vs T-ALL (right). A regression line is shown in blue, with a shaded gray band indicating the 95% confidence interval. The Pearson correlation coefficient (*r*) and its related *p* value (two-sided) for the relationship between DNA methylation and CTCF binding are shown at the top left. **g** Merged Hi-C contact map of the *KLF4* locus, comparing interactions between the CIMP (uppermost triangle, *n* = 5) and AML groups (bottom triangle, *n* = 5). ΔTADs are marked with a triangle in each half, and DIs are indicated with circles. Underneath, loops detected in this region are shown in black if they are invariable across conditions and in red or blue if they are gained or lost in CIMP relative to AML, respectively. The tracks below display MCIP-seq, CTCF ChIP-seq, and H3K27ac ChIP-seq from CIMP (*n* = 4) and AML (*n* = 4). Peaks gained in CIMP are highlighted in turquoise, whereas lost peaks are highlighted in light brown. The last track shows p300 binding measured by ChIP-seq in the K562 cell line. **h** Same as **g**, but the *CEBPD* locus is shown instead. **i**. Same as **g**, but the *GATA3* locus is shown.

**Whole exome sequencing.** The Genomic DNA Clean & Concentrator kit (ZYMO Research) was used to remove EDTA from the DNA samples. Sample libraries were prepared using 100 ng of input according to the KAPA HyperPlus Kit (Roche) using Unique Dual Index adapters (Integrated DNA Technologies, Inc.). Exomes were captured using the SeqCap EZ MedExome (Roche Nimblegen) according to SeqCap EZ HyperCap Library v1.0 Guide (Roche) with the xGen Universal blockers – TS Mix (Integrated DNA Technologies, Inc.). The amplified captured sample libraries were paired-end sequenced (2 × 100 bp) on the Novaseq 6000 platform (Illumina) and aligned to the hg19 reference genome using the Burrows-Wheeler Aligner (BWA)[82], v0.7.15-r1140.

**Chromatin immunoprecipitation followed by sequencing (ChIP-Seq).** ChIP-seq data with antibodies targeted at histone marks (H3K27ac, H3K27me3) were performed as described previously with slight modifications[83]. Briefly, cells were crosslinked with 1% formaldehyde for 10 minutes at room temperature, and the reaction was quenched with glycine at a final concentration of 0.125 M. Chromatin was sheared using the Covaris S220 focused-ultrasonicator to an average size of 250–350 bp. A total of 2.5 μg of antibody against H3K27ac (Abcam, ab4729) or H3K27me3 (Diagenode, C15410069) was added to sonicated chromatin of 2 × 10⁶ cells and incubated overnight at 4 °C. Protein A sepharose beads (GE Healthcare) were added to the ChIP reactions and incubated for 2 h at 4 °C. Beads were washed and chromatin was eluted. After crosslink reversal, RNase A, and proteinase K treatment, DNA was extracted with the Monarch PCR & DNA Cleanup kit (NEB). Sequencing libraries were prepared with the NEBNext Ultra II DNA Library Prep Kit for Illumina (NEB) according to the manufacturer's instructions. The quality of dsDNA libraries was analyzed using the High Sensitivity D1000 ScreenTape Kit (Agilent), and concentrations were assessed with the Qubit dsDNA HS Kit (Thermo Fisher Scientific). Libraries were either single-end sequenced on a HiSeq 3000 (Illumina) or paired-end sequenced on a Novaseq 6000 (Illumina).

ChIP-seq for CTCF, TCF7, CEBPA, and SPI1 (PU.1) was performed as follows. For TCF7 immunoprecipitation, cells were double crosslinked for 45 minutes at room temperature using 2 mM DSG (ThermoFisher Scientific, 20593) followed by a crosslink for 10 minutes at room temperature with 1% formaldehyde. When using antibodies against either CTCF, CEBPA, or SPI1, cells were single-crosslinked for 10 minutes at room temperature with 1% formaldehyde. All reactions were quenched with glycine at a final concentration of 0.125 M. When performing the TCF7 ChIP, cells were washed 3 times using cells lysis buffer A (10 mM Tris-HCL pH 7.5, 10 mM NaCl, 3 mM MgCl, 0.5% NP40). After last wash, cells were lysed in sonication lysis buffer (0.8% SDS, 160 mM NaCl, 10 mM Tris-HCL pH 7.5, 10 mM NaCl, 3 mM MgCL, 1 mM CaCl2, and 4% NP40). When performing the CEBPA or SPI1 ChIP, cells were lysed in lysis buffer B (50 mM Tris pH 8, 10 mM EDTA, 1% SDS). When performing the CTCF ChIP, cells were lysed in lysis buffer C (10 mM HEPES/NaOH, 85 mM NaCl, 1 mM EDTA) followed by a nuclear lysis buffer (50 mM Tris-HCl pH 7.5, 1% SDS, 0,5% EMPIGEN, 10 mM EDTA). All lysates were sonicated on a Bioruptor Pico sonication device (Diagenode). Chromatin was diluted 4 to 5 times using IP dilution buffer (1.1% Triton X-100, 0.01% SDS, 167 mM NaCl, 16.7 mM Tris-HCL pH 8.0 and 1.2 mM), except for the CTCF ChIP. A 2% sample was saved as a chromatin input control. A total of 2 ug of antibody against TCF7 (Cell Signaling 2203 S), 5 ug of antibody against CEBPA (Santa Cruz Biotechnology SC9314) or 0.4 ug of antibody against SPI1 (Cell Signaling 2266 S) was added to sonicated chromatin of 20 × 10⁶ cells or 4 ug of antibody against CTCF (Cell Signaling 2899 S) was added to sonicated chromatin of 2 × 10⁶ cells and incubated overnight at 4 °C. Chromatin bound antibody was precipitated with prot G Dynabeads (Thermo Fisher Scientific) and washed with low salt buffer (20 mM Tris pH 8, 2 mM EDTA, 1% Triton, 150 mM NaCl), high salt buffer (20 mM Tris pH 8, 2 mM EDTA, 1% Triton, 500 mM NaCl), LiCl buffer (10 mM Tris, 1 mM EDTA, 0.25 mM LiCl, 0.5% IGEPAL, 0.5% Sodium-Deoxycholate) and TE (10 mM Tris pH 8, 1 mM EDTA). In the TCF7 or CTCF ChIP chromatin was eluted in elution buffer A (0.1 M Sodiumhydrogencarbonate, 1% SDS). In the CEBPA or SPI1 ChIP chromatin was eluted in elution buffer B (25 mM Tris, 10 mM EDTA, 0.5% SDS). Crosslinks were reversed overnight at 65 °C in the presence of proteinase K (New England Biolabs P8107S), and DNA was extracted using a MinElute PCR Purification Kit (Qiagen 28004). Sequencing libraries were prepared using the MicroPlex Library v3 (Diagenode C05010001) preparation kit combined with dual indexes (Diagenode C05010008) and sequenced on the NovaSeq 6000 platform (Illumina).

ChIP-seq reads were aligned to the human reference genome build hg19 with either *bowtie*[75] (v1.1.1) for single-end data or bowtie2[84] (v2.3.4.1) for paired-end data. Bigwig files were generated for visualization with *deepTools bamCoverage*[76] (v3.5.1), and the options --normalizeUsing RPKM --smoothLength 100 --binSize 20. For data with narrow read distributions (H3K27ac, CTCF), peak calling was performed with *MACS2*[77] (v 2.2.7.1) using default settings, and the resulting peaks were filtered against the ENCODE blacklisted regions[78]. For H3K27me3, which is found in broad domains, peak calling was performed with EPIC2[85].

**Assay for transposase-accessible chromatin using sequencing (ATAC-seq).** ATAC-seq was essentially carried out as described[86]. Briefly, prior to transposition the viability of the cells was assessed, and 1 × 10⁶ cells were treated in a culture medium with DNase I (Sigma) at a final concentration of 200 U ml⁻¹ for 30 minutes at 37 °C. After Dnase I treatment, cells were washed twice with ice-cold PBS, and cell viability and the corresponding cell count were assessed. 5 × 10⁴ cells were aliquoted into a new tube and spun down at 500 × *g* for 5 minutes at 4 °C, before the supernatant was discarded completely. The cell pellet was resuspended in 50 μl of ATAC-RSB buffer (10 mM Tris-HCl pH 7.4, 10 mM NaCl, 3 mM MgCl2) containing 0.1% NP40, 0.1% Tween-20, and 1% Digitonin (Promega), and was incubated on ice for 3 minutes to lyse the cells. Lysis was washed out with 1 ml of ATAC-RSB buffer containing 0.1% Tween-20. Nuclei were pelleted at 500 × *g* for 10 minutes at 4 °C. The supernatant was discarded carefully, and the cell pellet was

resuspended in 50 μl of transposition mixture (25 μl 2× tagment DNA buffer, 2.5 μl transposase (100 nM final; Illumina), 16.5 μl PBS, 0.5 μl 1% digitonin, 0.5 μl 10% Tween-20, 5 μl H2O) by pipetting up and down six times. The reaction was incubated at 37 °C for 30 minutes with mixing before the DNA was purified using the Monarch PCR & DNA Cleanup Kit (NEB) according to the manufacturer's instructions. Purified DNA was eluted in 20 μl elution buffer (EB) and 10 μl purified sample was objected to a ten-cycle PCR amplification using Nextera i7- and i5-index primers (Illumina). Purification and size selection of the amplified DNA were carried out with Agencourt AMPure XP beads. For purification the sample to beads ratio was set to 1:1.8, whereas for size selection the ratio was set to 1:0.55. Purified samples were eluted in 15 μl of EB. The quality and concentration of the generated ATAC libraries were analyzed using the High Sensitivity D1000 ScreenTape Kit (Agilent) and libraries were sequenced paired-end on a NovaSeq 6000 (Illumina).

ATAC-seq reads were aligned to the human reference genome build hg19 with *bowtie2*[84] (v2.3.4.1), which is recommended for longer reads, and mitochondrial and duplicate reads were excluded. Bigwig files were generated as described above. Peak calling was also performed with MACS2[77] (v 2.2.7.1), but with the following settings: *--nomodel --shift 100 --extsize 200*. The resulting peaks were filtered against the ENCODE blacklisted regions[78].

**Hi-C**. Low input Hi-C was performed using 12k flow-sorted cells as previously reported[87]. The following procedural modifications was made: the mock PCR amplification was monitored by qPCR instead of by using an agarose gel. This was done by removing the magnetic beads and adding 20× sybr (Biotium 3100) to a small aliquot of the PCR reaction after two cycles. Amplified libraries were sequenced on a Novaseq 6000 instrument.

Hi-C data were first processed with *HiCUP*[88] v0.8.2, a pipeline for mapping and processing Hi-C data that removes technical artifacts and other invalid or uninformative di-tags. As part of this pipeline, the reads were aligned to the human reference genome build hg19 using *Bowtie2*[84] v2.3.4.1. Filtered di-tags were then extracted with the script *hicup2juicer* and subsequently binned with *juicer tools pre*[89] v1.22.01 at the default resolutions 2.5 Mb, 1 Mb, 500 Kb, 250 Kb, 100 Kb, 50 Kb, 25 Kb, 10 Kb, and 5 Kb. The resulting.hic files were used for visualization. TADs and loops were identified for each group of leukemias with the *findTADsAndLoops.pl find* script of the HOMER suite with the parameters –res 5000 and –window 10,000. Loops and TADs from each group were then combined into a single list with the *merge2Dbed.pl* script and individual scores were calculated per each sample with *findTADsAndLoops.pl score* using the same settings as above. These scores were imported to *DESeq2*[90] (v1.24.0) with the *DESeqDataSetFromMatrix* function and transformed with *varianceStabilizingTransformation* for unsupervised clustering analysis, as described above for other epigenomics data. Differences in TADs and loop scores between conditions were computed with a two-tailed Wald test using the *DESEq* function. The Benjamini−Hochberg method was applied to correct for multiple hypothesis testing with an FDR < 0.05. The results were visualized with *plotgardener*[91].

Because each dataset was sequenced at a relatively low depth of coverage (average = 398 M paired-end reads, 300 M valid pairs, and 155 M unique pairs), identification of 3D organization features was conducted in aggregate for each group and subsequently merged into a master list. The 4537 TADs and 9443 loops detected across all datasets were comparable to previous Hi-C results, such as 5975 domains and 6058 loops in K562 or 9274 domains and 9449 loops in GM12878[43]. Roughly 40-50% of CTCF sites overlapped with TAD boundaries and 10-20% with loop anchors, with minimal differences between variable and unchanged peaks (Supplementary Fig. 12a).

**Bioinformatics and data visualization**
Statistical tests were conducted on R version 4.1.0 unless otherwise specified. Most plots were generated using the *ggplot2* R package, whereas heatmaps were created with *ComplexHeatmap*[92] and genomic regions were visualized with *plotgardener*[91].

An extended description of the methods, with additional details on data generation and analyses, is provided in the Supplementary Information.

**Identification of functional regions.** Putative enhancer and promoter regions were defined for genome-wide quantification of methylation. In both cases, they were defined by three complementary criteria: (a) relative position to genes, (b) telltale histone marks, and c) eRNA expression. For enhancer identification, we constructed a consensus collection of H3K27ac-marked regions present in 3 or more samples from the CIMP, AML, T-ALL, and HSPC groups, excluding peaks that overlapped with 1 kb windows around transcriptional start sites (TSS) by at least 5% of their width. This list was intersected with a collection of open chromatin regions derived from the same groups (detectable in at least 3 samples) and with putative enhancers detected by CAGE-seq by the FANTOM consortium[93] (*human_permissive_enhancers_phase_1_and_2.bed*). For promoter identification, we downloaded the CAGE peaks assigned to TSS by the FANTOM consortium (*hg19.cage_peak_phase1and2combined_tpm_ann.osc.txt.gz*), excluded those not expressed in healthy hematopoietic cells, and intersected them with H3K4me3 peaks from CD34+ cells obtained from ENCODE[78].

CpG islands, identified according to the original criteria of Gardiner-Garden and Frommer[94], were downloaded from the UCSC browser.

For the comparison of genome-wide methylation levels at these functional regions, a Welch's t-test was used, which is quite robust to deviations from this distribution when sample sizes are very large, by virtue of the central limit theorem[95]. The effect size was calculated as Cohen's d (1), which standardizes the difference between two means by the pooled standard deviation:

$$Cohen's\ d = \frac{\bar{x}_1 - \bar{x}_2}{\sqrt{\frac{(s_1^2 + s_2^2)}{2}}}, \tag{1}$$

where $x$ is the mean of each group, and $s$ is the standard deviation

The result is the number of standard deviations units that separate two groups, with $D = 1$ indicating that the mean of one group is 1 standard deviation larger than that of the other. Typically, values below 0.2 are considered small and above 0.8 are considered large[96].

**Quantification and differential analysis of peak-based data.** Peak signal (MCIP-seq, ChIP-seq, ATAC-seq) was quantified with the *DiffBind* R package[97] as follows. First, all peaks were combined in a single master list using the default settings of the package, keeping only peaks present in at least 2 samples and removing chromosomes not present in the primary assembly and unassigned sequences. Only peaks with a −log(q value) higher than 10, as determined by MACS2, were considered. Overlapping peaks across multiple samples were combined into a single entry. The *dba.blacklist* function was used with the greylist argument set to false to remove only blacklisted regions. Then, reads mapping to this master list were counted for each sample, subtracting reads mapping to input DNA samples processed in the same way.

For differential analysis, data were normalized using the trimmed mean of the *M* values (TMM)[98] with the sum of reads in consensus peaks as the library size (argument normalize=DBA_NORM_TMM in *DiffBind*). These normalization factors and the raw counts were passed to DESeq2 (v1.34.0) with the *dba.analyze* command and differential regions were identified as those with a false discovery rate (FDR) < 0.5 by the Benjamini−Hochberg method[90]. Peaks were annotated to the closest gene with the *ChIPpeakAnno* package[99].

Identification of transcription factor binding sites enriched at peak sets was conducted using *LOLA*[100]. For visualization, results were aggregated by TF, selecting the dataset with the lowest *p* value from a hematopoietic tissue, if possible, or another tissue otherwise.

**Clustering of transcriptional and epigenomics data.** MCIP-seq, RNA-seq, ATAC-seq, and CHIP-seq data were processed similarly to identify relevant relationships between CIMP leukemias and other diseases. Briefly, raw counts were imported to DESeq2 with the *DESeqDataSetFromMatrix* function and transformed with *varianceStabilizingTransformation* to reduce the dependence of the variance from the mean. The 5000 regions or genes with the highest variance in transformed counts were selected for further analysis. PCA, multi-dimensional scaling (MDS), t-distributed Stochastic Neighbor Embedding (t-SNE), and UMAP were used for dimensionality reduction and visualized with *ggplot2*. Since results were highly comparable across different strategies (Supplementary Fig. 2), only PCA and UMAP were used in the rest of the figures. Moreover, heatmaps of either Pearson correlation or Euclidean distances between samples were created with *ComplexHeatmap*[92].

In the clustering analyses of ATAC-seq data, patient UKR021 was excluded due to excessively low quality (hence, *n* = 81). However, it was included in other analyses performed with those data.

**Analysis of MethylationEPIC array data.** Illumina Infinium MethylationEPIC datasets compiled from CIMP AML, T-ALL[25] (GSE147667), and AML[26] (GSE159907) were subjected to quality control, preprocessing, and normalization using *RnBeads*[101]. First, we filtered out SNP-overlapping probes, cross-reactive probes, sites outside of CpG context, and those mapping to sex chromosomes, as well as probes with detection *p* value > 0.05, and sites covered by fewer than three beads. Next, beta-values were normalized and background subtraction was performed using the *scaling.internal* and *sesame.noobsb* options.

For comparison of global methylation at different genomic regions, CpG islands were defined according to *RnBeads* annotation. Illumina MethylationEPIC annotation files were used to map CpG sites to FANTOM enhancers[93] and gene bodies, and to match CpG sites to their associated genes. Promoters were defined as the region 1500 bp upstream and 500 bp downstream of transcription start sites.

Differential methylation between groups was computed at the level of CpG sites using RnBeads. CpG sites with absolute mean beta value difference > 0.2 and FDR-adjusted *p* value < 0.05 were considered differentially methylated. Among differentially methylated sites, enrichments of transcription factor binding sites and gene ontologies were calculated using the LOLA[100] and clusterProfiler R[102] packages, respectively.

*Salmon*[103] was used to quantify the expression of individual transcripts, which were subsequently aggregated to estimate gene-level abundances with the R package *tximport*[104]. Human gene annotation derived from GENCODE[105] v30 was downloaded as a GTF file and used for the quantification. Both gene- and transcript-level abundances were normalized to counts per million for visualization in the figures of this paper. Differential gene expression analysis of count estimates from Salmon was performed with *DESEq2*[90] v1.34.0. The Benjamini–Hochberg method was applied to correct for multiple hypothesis testing with an FDR < 0.05.

**Integration of MCIP-seq and RNA-seq data.** To determine how changes in methylation relate to gene expression, we quantified MCIP-seq signal exclusively at promoters, defined on the basis of H3K4me1 peaks as explained above. Then, we computed differential methylation with *DiffBind* and we left-joined the results with the differential expression previously calculated with *DESeq2* using gene names to match the records. The results were visualized as a Starbust plot, in which the -log10(FDR) multiplied by the sign of the fold change are

shown for the MCIP-seq data in the *X* axis and the RNA-seq data in the *Y* axis. To facilitate the identification of TFs involved in hematopoiesis (those within the GO term GO:0030097), they were highlighted with red circles and labeled.

**Identification and analysis of small genetic variants.** Single nucleotide variant (SNV) and small insertion/deletion (indel) detection was performed with a custom script that integrated variants called by multiple software tools, including HaplotypeCaller and *MuTecT2* from GATK v4.0.0[106], *VarScan2*[107], *bcftools*[108], *Strelka2*[109] and *Pindel*[110]. A highly optimized in-house tool (*annotateBamStatistics*, available at https://gitlab.com/erasmusmc-hematology/annotatebamstatistics) was then used to compute the variant allele frequency (VAF) of every variant as well as position-specific metrics for such as strand bias, number of clipped reads or the number of alternative alignments (Supplementary Data 2). The combined list of variants was subjected to stringent filtering to remove low-quality positions, considering the following criteria:

a. strand bias between 0 and 1 for regions within the exome capture (+200 bp)
b. total sequencing depth of at least 8 reads and 4 for the variant allele
c. alignment quality 40 or more and base calling score 30 or more
d. fewer than 40% of reads mapping to a base other than the reference and alternative alleles
e. maximum of 10% of the reads with an alternative alignment or a superior alternative alignment score in *bwa* (XS)
f. removal of extremely long indels (500 bp or more)
g. removal of variants in simple repeats as detected by RepeatMasker[111] (downloaded from UCSC)
h. removal of variants in highly repetitive genomic regions, as determined by 95% or more identity to another region in selfChain link files from UCSC
i. removal of clusters of at least 3 SNVs with a distance of less than 5 bp from each other

Furthermore, since we did not have control material for these patients, we selected mutations likely to be somatic among the variants identified by WES based on functional annotation by Annovar[112]. Thus, we first considered mutations complying with the following criteria: a) located in exons or in splicing acceptor regions, b) non-synonymous SNV or indels, and c) with a VAF of at least 1%. Single nucleotide polymorphisms (SNPs) with a population frequency higher than 0.0002 were excluded unless they were reported in the COSMIC database v94[113] in at least 5 hematological cancers, or they were present in genes with frequent clonal hematopoiesis mutations (*DNMT3A, TET2, ASXL1*)[114]. Variants present in a healthy donor (though not a paired matched control) were also removed to further eliminate common variants and technical artifacts. Moreover, variants present in a blacklist of frequent non-somatic variants found in WES from AML and CD34+ cells were discarded. Finally, probable oncogenic variants were selected as those that fulfilled one or more of the following conditions: (i) in the COSMIC database; (ii) frameshift, stopgain or startloss; (iii) majority of damaging functional predictions by tools such as PolyPhen, SIFT, LRT and others.

Given the difficult interpretation of some of these variants, the resulting list was further reduced by selecting only genes previously reported in leukemia (Disgenet database[115]), cancer (COSMIC[113]), or relevant in hematopoiesis (GO term GO:0030097). This file (Supplementary Data 2) was used as an input for the *oncoPrint* function of the *ComplexHeatmap* R package to show the distribution of mutations in this cohort.

To compare the mutational landscape of CIMP leukemias to that of other acute leukemias, we compiled data from published studies on AML[27,116], T-ALL[2,117–119], ETP-ALL[23,118–121] and T/M MPAL[24,122]. We then

counted the occurrences of mutations in every gene and conducted a two-tailed Fisher test.

**Copy number alteration detection.** Copy number analysis on WES data was performed with *CNVkit*[123] v0.9.9 in two steps. First, a pooled reference was generated based on 12 datasets from healthy CD34+ cells (9 from adult bone marrow and 3 from cord blood). As suggested by the instructions of the program, 5 kb regions of poor mappability were excluded from the analysis. Subsequently, the reference was employed to compute log2 copy ratios and infer discrete copy number segments using the default settings of *CNVkit*. Finally, we derived absolute integer copy numbers of these segments with the function *cnvkit call* and copy number alterations (CNAs) were computed at the gene-level with *cnvkit genemetrics*. Copy number data were summarized across all AML samples and represented as a heatmap with *ComplexHeatmap*. Scatter plots of specific regions such as *NF1* were created with *cnvkit scatter*.

These results were validated by orthogonal analyses with *CNV Radar*[124] on WES data and *Control-FREEC*[125] on input DNA sequencing data generated for the ChIP-seq and MCIp-seq experiments. For *CNV Radar*, common SNPs (db SNP 151) were annotated in the variants called by *bcftools call* with the *SnpSift*[126] tool, as prescribed by the instruction manual. This step ensures that the B-allele frequency (BAF) is only calculated with polymorphisms that are expected to be heterozygous, avoiding distortions introduced by potentially subclonal somatic mutations. A panel of non-matched normals was used as a control analogously to the previous analysis with *CNVkit*. *Control-FREEC* was run without controls in large windows of 100,000 bp to compensate for the low sequencing depth of the files.

**Fusion gene detection.** Fusion gene identification was carried out on RNA-seq reads by means of an ensemble of software tools, namely *STAR-Fusion*[127], *FusionCatcher*[128], *Arriba*[129], *Pizzly*[130], *JAFFA*[131] and *SQUID*[132]. Results from these tools were integrated with *fusion-reporter*, a Python script developed for the *nf-core* framework of bioinformatics pipelines[133]. Fusion gene candidates were filtered with the databases bundled with FusionCatcher, thereby excluding those previously found in studies of healthy tissues or involving partners in close proximity. Majority voting by a minimum of 3 tools was employed to select the final fusion candidates per sample, which were then combined into a single master list.

The combined list of fusions was further annotated based on their presence in fusion gene databases (FusionGDB, COSMIC, and Mitelman) or previous reports of that fusion in leukemia studies[118,134–136]. Fusions whose individual genes are involved in leukemia according to the Disgenet database[115] were also annotated. The master list and the leukemia-related annotations were visually represented with the *oncoPrint* function of the *ComplexHeatmap* R package.

**Gene set enrichment analysis and identification of hematopoietic signatures.** Pre-ranked gene set enrichment analysis (GSEA)[137] was computed with the *fgsea* R package using the multilevel splitting Monte Carlo approach to calculate *p* values, with the settings minSize=15, maxSize=5000. For every comparison, the input genes were ranked by the -log10(FDR) multiplied by the sign of the fold change, both obtained from DESeq2. We used the MSigDB C5 collection, containing GO terms, to investigate enrichment for gene functions and biological processes[138]. We also employed a customized MSigDB C2 collection, containing version v7.5.1 of C2 plus several hematopoiesis-related datasets kindly provided by Dr Charles Mullighan. Moreover, we added datasets derived from supervised comparisons between ETP-ALL and other T-ALLs[22,23], as well as a signature of leukemia induced in DN2 thymocytes mice by a retrovirus coexpressing *Myc* and *Bcl2*[139]. Both C2 and C5 were downloaded with the *msigdbr* R package.

To evaluate the potential cell of origin of CIMP leukemias, we analyzed the samples with single sample GSEA (ssGSEA) implemented as a part of the *GSVA* R package[140]. For this analysis we focused on gene sets specific for certain hematopoietic cell fractions, obtained from[141] and[142]. With that same goal, we employed *CIBERSORTx*[30], originally designed to dissect cell type proportions in a mixture on the basis of a signature matrix. Signature matrices were generated from a single-cell RNA-seq dataset obtained from the Atlas of Human Blood Cells[143].

**Integration of ATAC-seq and RNA-seq data.** To characterize the transcriptional consequences of epigenetic changes in CIMP leukemias beyond promoter hypermethylation, we integrated chromatin accessibility with gene expression data. First, we classified ATAC-seq peaks into putative promoters if they overlapped with a region of 500 bp around a TSS ± 500 bp, and the rest as putative enhancers. In order to assign putative enhancers to their target genes, we integrated multiple layers of information, namely: distance between enhancer and the nearest promoter, correlation between ATAC-seq peaks, co-occurrence of both elements in the same TAD, contact by a loop, previous assignment by the GeneHancer database[144]. Next, we used *DiffBind* to compute differential ATAC-seq signal between putative enhancers successfully assigned to a target gene. For each of these genes, differences in the accessibility of their enhancers were linked to differences in their expression. The results of this analysis were visualized as a Starbust plot as described for RNA-seq and MCIP-seq integration in the corresponding section of the Methods.

**Analysis of motif activity in chromatin accessibility data.** The regulatory networks driving a differentiation block in CIMP were investigated with *chromVAR* (v1.20.2), a tool that estimates TF motif activity by computing bias-corrected deviations in chromatin accessibility at motif-containing peaks relative to the expectation[145]. First, motifs for human TFs were downloaded from *JASPAR* (v2022)[146] and filtered to exclude any genes not expressed in any leukemia samples of our RNA-seq cohort with at least 1 TPM, yielding a total of 721 motifs. The consensus peaks derived from ATAC-seq data (see "Quantification and differential analysis of peak-based data") were then evaluated for matches with any of those motifs using the function *matchMotifs* from the *motifmatchr* package (v1.20.0). Having determined which peaks contained which motifs, genome-wide deviations in motif accessibility were calculated with the *computeDeviations* using the ATAC-seq computed by *DiffBind*. Variability in the activity of each motif, defined as the standard deviation of the Z-scores across the entire cohort, was computed with *computeVariability*. Differential accessibility was calculated as the difference in the mean of deviation Z-scores, and statistical significance was assessed with a two-tailed Wilcoxon test. The results of this analysis were depicted as a volcano plot.

Putative positive regulators were identified by calculating the correlation between the deviations estimated by *chromVAR* and the expression of the gene in the same sample, expressed in TPM. This is an approach frequently used in single-cell data analysis and it relies on the assumption that the binding sites of activators become accessible when such genes are expressed[147–150].

Furthermore, we conducted TF footprinting with *TOBIAS*[151], a software suite specifically designed for the analysis of ATAC-seq data. Since the high depth of coverage is recommended for footprinting and *TOBIAS* does not deal with replicates, we first used *samtools merge* to aggregate multiple samples from each group: all available CIMP cases (n = 9) and a representative fraction of AML (n = 20) and T-ALL (n = 20). We called peaks on each combined BAM file as described before and, for each comparison, we merged the peak files of the two conditions involved. Then, we ran *TOBIAS ATACorrect* to generate bigwig tracks corrected for the Tn5 sequence bias of ATAC-seq, followed *by TOBIAS FootprintScores* to calculate a footprinting score based on the depletion of signal relative to nearby regions. Finally, *TOBIAS BINDetect*

combined these footprint scores with the information of TF binding motifs derived from *JASPAR* (filtered as described above). The results of this analysis were depicted as a volcano plot. Moreover, *TOBIAS PlotAggregate* was used to visualize the ATAC aggregated signal of selected TF motifs in various conditions of interest.

**Integration of Hi-C data with gene expression and CTCF binding.** TAD boundaries were defined as 5000 bp regions (same as the resolution used for TAD calling) centered on their borders. CTCF binding sites overlapping with those regions were identified, but only a single peak with the smallest FDR was kept for each boundary, depending on which comparison was conducted. Similarly, MCIP-seq peaks overlapping with boundaries were selected based on their FDR for each comparison. Differential expression of genes within TADs was also incorporated. This information is summarized in Supplementary Data 40, which was used to identify variable TADs with a) significant changes in CTCF binding in their boundaries, and b) differentially expressed genes.

Loop anchors were defined according to the coordinates provided by *HOMER*, with an extra padding of 5000 bp on each side to account for the resolution used in loop calling. Enhancers and promoters (see "Identification of functional regions") in the vicinity of loop anchors were identified at a distance of 25,000 bp or less. Thus, we could select enhancer-promoter loops as those with an anchor close to an enhancer and a promoter on each side (Supplementary Data 41). For loop anchors attached to a promoter, differential expression of the corresponding gene between the conditions of interest was also retrieved.

**Statistics and reproducibility**
Unless otherwise specified, analyses performed in this study used the sample sizes shown at the bottom of Supplementary Data 1. No sample size calculation was performed, as the sample size was determined by the availability of samples and/or data. However, sample sizes for all the main phenotypic groups studied here (CIMP, AML, T-ALL, and CD34+) were large enough to robustly identify statistically significant differences. Briefly, we collected samples from 376 leukemia patients and 19 healthy donors in total, although not all data were generated for every individual. CIMP cases ($n = 14$) were the same across all datasets, but the composition of the T-ALL, AML, and CD34+ groups varied. For AML, a cohort representative of recurrent genetic abnormalities was used ($n = 224$). RNA-seq was previously available from another study[152], but MCIP-seq, ChIP-seq, ATAC-seq, and Hi-C data were generated ad hoc for a fraction of those patients. Similarly, RNA-seq of T-ALL patients ($n = 114$) was available from another ongoing study, but it was supplemented with additional epigenomics data. The T-ALL patients sequenced with MCIP-seq were not included in any other experiment. ATAC-seq from AML patient UKR201 was removed from the clustering analysis due to low quality, but not from other analyses.

Every measurement of the same type was taken from a different sample, i.e., no technical replicates were produced. The reproducibility of the findings was assessed by incorporating orthogonal evidence. For example, publicly available MethylationEPIC array data from T-ALL[25] and AML[26] were used to confirm that CIMP patients exhibit strong methylation signatures similar to those of ETP-ALL. Randomization was not applicable to the current study because CIMP leukemias were compared to those without this phenotype. Patients included in control groups (AML, T-ALL) were selected in such a way that they reflected the variety of mutational backgrounds in those leukemias. Donors for CD34+ cells were randomly selected.

The statistical tests used in the analysis of these data are described in the corresponding section of the Methods and, when appropriate, in the legends of the visual elements where the results of these analyses are presented. Generally, we used a two-sided Wald test for count data

derived from omics techniques; a two-sided Wilcoxon test for pairwise comparisons of other data, such as average methylation levels at different genomics features; and a Fisher's exact test for enrichment (one-sided) and association (two-sided) analyses.

**Reporting summary**
Further information on research design is available in the Nature Portfolio Reporting Summary linked to this article.

## Data availability
The raw RNA-seq data of AML patients have been previously used in another study[152] and are available at the European Genome-phenome Archive (EGA) under accession number EGAS00001004684. All the other raw sequencing data derived from donors or patients have been generated in this study and are deposited at the EGA under accession number EGAS00001007094. This EGA study includes the following datasets: MCIP-seq, RNA-seq, ATAC-seq, ChIP-seq (H3K27ac, CTCF, SPI1, CEBPA, TCF7), and Hi-C. Since these data are derived from human subjects, they are only available under restricted access, which can be requested for each dataset separately on the EGA website. Requestors must sign a data access agreement outlining the terms and conditions for data use and fill in a form specifying their research question. Requests will be processed within 1 week, and the data will be available for a maximum of 2 years unless an appeal for extension is submitted. Processed data are publicly available in ArrayExpress with the following identifiers: E-MTAB-13117 (CTCF ChIP-seq), E-MTAB-13118 (ATAC-seq), E-MTAB-13119 (H3K27ac ChIP-seq), E-MTAB-13120 (MCIP-seq), E-MTAB-13121 (RNA-seq), E-MTAB-13122 (Hi-C), E-MTAB-14060 (TF ChIP-seq). The remaining data are available within the Article, Supplementary Information, and Source Data that accompany this article. In addition, we have used publicly available data from the ENCODE[78] and FANTOM[93] consortia, as well as a single-cell RNA-seq dataset of hematopoietic cells obtained from the Gene Expression Omnibus (GEO) database under the accession code GSE149938[143]. We also used Illumina Infinium MethylationEPIC data from T-ALL[25] (GSE147667) and AML[26] (GSE159907). Source data are provided with this paper.

## Code availability
All software tools employed in this study are freely or commercially available (see Methods). R code used in the analysis of the data presented here can be found in Supplementary Code 1.

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

## Acknowledgements

We would like to thank the colleagues from the bone marrow trans-plantation group and the molecular diagnostics laboratory of the department of Hematology (E. Braakman, P.J.M. Valk) as well as colla-borators from the department of Clinical Genetics (H.B. Beverloo) at the Erasmus University Medical Center for storage of samples, molecular and cytogenetic analysis of the leukemia cells. We are also grateful to Remco Hoogenboezem for assistance in developing bioinformatics tools and to the rest of our colleagues in the department of Hematology for input during work discussions. Furthermore, we thank Roberto Avellino for critically reading the manuscript. This work was funded by grants from the following organizations: (1) Dutch Cancer Foundation: EMCR 2015-7935 (R.D.), EMCR 2015-7550 (B.W.); (2) Leukemia & Lym-phoma Society (LLS), Special Fellowship in Clinical Research: Grant # 4317-16 (B.W.); (3) Deutsche Forschungsgemeinschaft SPP2202 Priority Program: GE 202/1-1 (C.G. and J.M.V.); (4) Medical Research Council: UK MC_UP_1605/10 (J.M.V.); (5) the Academy of Medical Sciences and the Department of Business, Energy and Industrial Strategy: APR3\1017 (J.M.V.); (6) German Cancer Aid (M.R.); (7) Wilhelm Sander Stiftung (M.R.); (8) Krebshilfe Antrag 111602 (M.R.).

## Author contributions

The study was designed and written by R.M.-L., B.W, C.G., R.D., and M.R. Wet lab experiments were performed by S. van H., M.N., C.G., A.S., N.D., C.E-V., L.S.-P., H.S, D.G., U.A., S.P, J.R., and A.F. Bioinformatical analyses were conducted by R.M-L., K.K., J.M.V., C.G., and M.R. Patient samples and data were provided by A.S., A.R., C.T., C.P., C.G., R.D., and B.W.

## Competing interests

The authors declare no competing interests.

## Additional information

**Roger Mulet-Lazaro** [1,2], **Stanley van Herk** [1,2], **Margit Nuetzel**[3], **Aniko Sijs-Szabo**[1], **Noelia Díaz** [4,5], **Katherine Kelly**[6], **Claudia Erpelinck-Verschueren** [1,2], **Lucia Schwarzfischer-Pfeilschifter**[3], **Hanna Stanewsky**[3], **Ute Ackermann**[3], **Dagmar Glatz**[3], **Johanna Raithel**[3], **Alexander Fischer** [3], **Sandra Pohl**[3,7], **Anita Rijneveld**[1], **Juan M. Vaquerizas** [4,8,9], **Christian Thiede**[10], **Christoph Plass**[6], **Bas J. Wouters** [1,2,12] ✉, **Ruud Delwel** [1,2,12] ✉, **Michael Rehli** [3,11,12] ✉ & **Claudia Gebhard**[3,11,12] ✉

[1]Department of Hematology, Erasmus MC Cancer Institute, Rotterdam, the Netherlands. [2]Oncode Institute, Utrecht, the Netherlands. [3]Department of Internal Medicine III, University Hospital Regensburg, Regensburg, Germany. [4]Max Planck Institute for Molecular Biomedicine, Muenster, Germany. [5]Renewable Marine Resources Department, Institute of Marine Sciences (ICM-CSIC), Barcelona, Spain. [6]Division of Cancer Epigenomics, German Cancer Research Center (DKFZ), Heidelberg, Germany. [7]Department of Conservative Dentistry and Periodontology, University Hospital Regensburg, Regensburg, Germany. [8]MRC London Institute of Medical Sciences, London, United Kingdom. [9]Institute of Clinical Sciences, Faculty of Medicine, Imperial College London, Hammersmith Hospital 8 Campus, London, United Kingdom. [10]Medizinische Klinik und Poliklinik I, Universitätsklinikum Carl Gustav Carus, Dresden, Germany. [11]Leibniz Institute for Immunotherapy (LIT), Regensburg, Germany. [12]These authors contributed equally: Bas J. Wouters, Ruud Delwel, Michael Rehli, Claudia Gebhard. ✉e-mail: b.wouters@erasmusmc.nl; h.delwel@erasmusmc.nl; michael.rehli@klinik.uni-regensburg.de; claudia.gebhard@klinik.uni-regensburg.de

