## [Peer Review File · Nature Communications]

Epigenetic alterations affecting hematopoietic regulatory networks as drivers of mixed myeloid/lymphoid leukemiaReviewers' Comments:

Reviewer #1:

Remarks to the Author:

The paper by Mulet-Lazaro et al. reports a detailed epigenetic analysis of mixed myeloid/lymphoid leukemia cells. The authors use a set of state-of-the-art techniques and suitable data analysis to identify specific epigenetic features of these mixed lineage leukemias. This revealed increased DNA methylation at genes encoding myeloid associated transcription factors, general hypermethylation and changes in CTCF binding and higher order chromatin states as compared to "classical" AML cells. Despite the similarities in the epigenetic landscape, WGS did not reveal any specific genetic alteration that could be linked to the phenotype of the cells.

The paper is well written, logical and presents a large amount of most interesting data. While the phenotypic features of the mixed lineage leukemias are well characterized by this report, the lack of functional analysis makes the paper somewhat descriptive. Increased methylation of transcription factor encoding genes associated with decreased transcriptional activity, may be a primary cause of the mixed phenotype as well as the differentiation block, however, as these ideas are not experimentally validated, they remain rather hypothetical.

Specific comments:

1: I find it difficult to understand the authors conclusion that the cell of origin for the CIMP would be a lymphoid restricted progenitor. The analysis in 2F suggest an at least as strong relationship to MEPs as to the CLP. A better description of how this conclusion was reached would be most useful.

2: It is difficult to interpret that data in figure 3A as not statistical analysis is presented.

3: The authors suggest that the silencing of transcription factors may be the cause of mixed phenotype as well as the differentiation block. While the latter is easily understood, it is not as easy to understand how silencing of activating transcription factors would result in aberrant gene expression. The discussion could be developed to address this subject in more detail.

Reviewer #2:

Remarks to the Author:

The work by Mulet-Lazaro et al. and colleagues uses orthogonal functional genomics assays to characterize the genomic, epigenomic and transcriptomic landscape of mixed myeloid/lymphoid leukemia. The authors identified that epigenetic dysregulation, rather than genetic alterations, drive mixed lineage leukemia. Specifically, mixed lineage leukemias appear to harbor a hybrid epigenetic landscape, with methylation patterns consistent with T-ALL, but a regulatory landscape (H3K27ac and ATAC-seq) more consistent with AML. Overall, this manuscript provided interesting insights on mixed lineage leukemias. One prominent drawback was the descriptive nature of the work, with no functional validation or mechanistic insights beyond genomic annotation. Key points the authors should address include:

1. In Figure 1A, is it possible to add T-ALL data to show closer relationship of CIMP leukemias with T-ALL versus AML? This is shown with UMAP analysis in Figure 1C but would also be relevant for Figure 1A.

2. In Figure 1C, several T-ALL samples cluster with AML for MCIP-seq, and several cluster with both AML and CIMP in the H3K27ac and ATAC-seq analyses. Are these ETP-ALL samples? Are they the same samples across the panels? The authors should clarify this. If these are indeed ETP-ALL samples, they should be distinguished from the remaining T-ALL samples on the figure panels.

3. Because the authors perform ATAC-seq, an analysis of TF footprints should be performed to identify candidate bound TFs at sites of interest. Moreover, comparisons of TF footprint intensity between CIMP vs AML or T-ALL can further identify key TFs that show differential activity. These comparisons can be performed at a subset of sites of interest using computational tools such as TOBIAS.

4. In Figure 2A, if the authors performed clustering analysis of methylation and epigenomic signatures in just the CIMP leukemias, is any genetic lesion(s) shared/common among samples that cluster together?

5. For the transcriptional analysis, the authors should try to integrate these data with their epigenomic profiling. For examples, do the authors identify any correlation in epigenetic changes at regulatory sites (H3K27ac/ATAC) between CIMP vs ETP-ALL or CIMP vs AML and the corresponding transcriptional effects on associated genes between CIMP vs ETP-ALL or CIMP vs AML? For instance, does reduced/enhanced H3K27ac signal in CIMP vs AML lead to reduced/enhanced transcription of a neighboring gene in CIMP vs AML? Of course, assigning a target gene to regulatory elements can be difficult and prone to error. However, I would expect some level of correlation to further demonstrate epigenetic changes in CIMP have functional consequences on transcription.

6. The authors should perform ChIP-seq for some key TFs that they claim are altered in CIMP versus AML and T-ALL samples (e.g., CEBPA, KLF4, RUNX1, etc.). This can also be performed in T-ALL or AML cell lines and the sites compared to the corresponding sites in patient samples exhibiting altered epigenetic state in CIMP (or reduced TF footprinting; see comment 3 above). Some of these TF profiles may already be available from ENCODE in related cell types.

7. Can the authors functionally demonstrate that their work provides therapeutic insights as CIMP leukemias overall have a poor prognosis? For instance, as the authors suggest, would NOTCH1-targeting drugs or decitabine be a viable treatment in CIMP PDXs? Are CIMP PDXs/patient samples more sensitive to NOTCH1-targeting drugs/decitabine compared to AML PDXs/samples? This experimentation would provide translational value and validation to an otherwise descriptive manuscript.

Reviewer #3:

Remarks to the Author:

The study by Mulet-Lazaro et al entitled title 'Epigenetic alterations affecting hematopoietic regulatory networks as drivers of mixed myeloid/lymphoid leukemia' used a multi-omic approach to evaluate the role of the epigenetic landscape in leukemias with ambiguous lineage. Of particular focus is a hypermethylated AML subtype which they labelled CIMP. This labeling is unfortunately confusing since CIMP subtyping is also used for subgrouping of e.g. T-cell leukemias (T-ALL CIMP negative and CIMP positive). In this manuscript, both the CIMP AML group (included in most figures) and the CIMP T-ALL subgroups are analyzed (Figure S2B). To avoid this confusion, a more specific naming of the hypermethylated AML subtype would have helped. The overall aim is relevant but unfortunately, the methods and the results (figures/tables) are not presented clearly in their current form.

Major comments:

✕ Even though the number of main figures is within the allowed number (10 figures/tables), the number of individual plots is extremely high (n=36 individual main figures and n=86 individual Supplementary figures, along with 38 supplementary tables). Some figures are shown both as main figures and supplementary figures (e.g. Fig 1C, Fig S2C). Not all figures and tables are referred to in the text and those one should be removed. There are results presented in the discussion section that have not been presented in the result section (Figure S12A-B), and they are recommended to be added to the result section.

- × The method description in the main text is very short and do not declare all method used (e.g. not the array data), and the patient material needs to be described in more detail (age, numbers, etc.). An extended description of the material and methods is needed, and are currently not included as Supplementary Information as claimed in the manuscript (row 430-431). These method details are necessary to evaluate the results and conclusions of the manuscript.
- × The Supplementary Figures lack figure legends and cannot be interpreted without this information. Several shortenings in the figure axis are not explained in the figure or text.
- × A descriptive title describing what the tables show would have been helpful.
- × Several figures lack information on the number of samples included in each analysis, some lack statistics and a description of subgrouping definitions (e.g. mutation subgrouping) making it difficult to interpret the results and significance of the data.

Reviewer #4:

Remarks to the Author:

RESPONSE TO REVIEWERS

We appreciate all the constructive comments and suggestions from the reviewers. We have systematically addressed all the comments point by point below.

Reviewer #1, expertise in leukemia, gene regulation and epigenetics (Remarks to the Author):

The paper by Mulet-Lazaro et al. reports a detailed epigenetic analysis of mixed myeloid/lymphoid leukemia cells. The authors use a set of state-of-the-art techniques and suitable data analysis to identify specific epigenetic features of these mixed lineage leukemias. This revealed increased DNA methylation at genes encoding myeloid associated transcription factors, general hypermethylation and changes in CTCF binding and higher order chromatin states as compared to “classical” AML cells. Despite the similarities in the epigenetic landscape, WGS did not reveal any specific genetic alteration that could be linked to the phenotype of the cells.

The paper is well written, logical and presents a large amount of most interesting data. While the phenotypic features of the mixed lineage leukemias are well characterized by this report, the lack of functional analysis makes the paper somewhat descriptive. Increased methylation of transcription factor encoding genes an associated decreased transcriptional activity, may be a primary cause of the mixed phenotype as well as the differentiation block, however, as these ideas are not experimentally validated, they remain rather hypothetical.

Specific comments:

1: I find it difficult to understand the authors conclusion that the cell of origin for the CIMP would be a lymphoid restricted progenitor. The analysis in 2F suggest an at least as strong relationship to MEPs as to the CLP. A better description of how this conclusion was reached would be most useful.

We thank the reviewer for these insightful remarks. We acknowledge that our conclusion about the cell of origin of CIMP leukemias in page 4 is not properly justified in the text. This section of the results, in itself, points to an early progenitor as a likely cell of origin, but indeed it falls short of determining whether said progenitor would be restricted to the lymphoid lineage. However, there are additional lines of evidence that we have taken into consideration to infer a lymphoid priming of the cell of origin, namely epigenetic similarity and mutational profiles. We have modified both the Results (page 4, lines 160-164) and the Discussion (page 9, lines 345-346) sections of the manuscript to better reflect the criteria we have used to reach this conclusion, which we also describe in more detail below.

- a) Transcriptional signatures: we used several methods to investigate whether transcriptional signatures of well-defined hematopoietic cell types are particularly enriched in CIMP leukemias. Gene set enrichment analysis (GSEA) revealed enrichment for T-cell gene sets compared to AML, with depletion of ETP and myeloid gene sets (Figure 2E, but also Tables S10 and S11). Conversely, ETP and GMP gene sets were more expressed in CIMP than in T-ALL, whereas T-cell signatures were depleted. Interestingly, HSC gene sets were enriched in CIMP relative to both AML and T-ALL, which again pointed to a more primitive cell of origin. Since GSEA is very dependent on the baseline for differential expression, we also compared CIMPs to HSPCs (Figure 2E, bottom panel, and Table S12). This analysis identified preferential activation of lymphoid gene sets and downregulation of genes associated with myeloid progenitors, hinting at a tilt towards the lymphoid lineage. It should be noted that MEP gene sets (like “Dick_MEP100”) were, in fact, downregulated in CIMPs.

However, this result could still be affected by the composition of the HSPC compartment, so we employed CIBERTSORTx to deconvolve the gene expression profiles of each leukemia into their constituent signatures based on publicly available single cell data (Newman *et al.*, 2019). We presented the results of this analysis as a heatmap in Figure 2F because it summarizes the key signatures that distinguish CIMP from other leukemias, but the Z-score normalization can be misleading because it a) magnifies differences

that are not significant, b) it hides the real scale of the data. In Figure S4C, we have shown the same data as a boxplot, which reveals that MEP signatures are, in fact, very similar across all leukemias, albeit slightly higher in CIMP than in AML ($p = 0.015$). On the other hand, it is clear that CIMPs are much more lymphoid (CLP signature) than AMLs, but also much more myeloid (GMP signature) than T-ALLs. Single sample GSEA showed largely concordant results (Figure S4E).

The interpretation of these analyses is not straightforward, and it is further hindered by the heterogeneity of the disease groups under consideration. Based on these results alone, a reasonable conclusion is that CIMPs have a transcriptional profile reminiscent of an early hematopoietic progenitor, but it is indeed hard to establish whether it is more myeloid or more lymphoid. That said, the fact that standard GSEA identifies activated lymphoid pathways in CIMPs relative to HSPCs could indicate a bias towards the lymphoid lineage. Moreover, the MLP signature was very high for CIMP leukemias in the single sample GSEA. Nonetheless, we realize that these weak transcriptional signatures are not sufficient to conclusively establish the cell of origin.

- b) Epigenetic similarity: dimensionality reduction and hierarchical clustering both showed strong similarity between CIMPs and T-ALL at the DNA methylation level, whereas gene expression was more ambiguous and ATAC-seq and H3K27ac profiles were closer to those of AML (Figure 1C, Figure S2D-E). Despite these partially contradictory results, it can be argued that DNA methylation is the trait most strongly associated with cell identity, as it is reliably transmitted during DNA replication and thus maintained across generations (Kim and Costello, 2017). During differentiation, hematopoietic cells undergo microwaves of *de novo* methylation and demethylation, but some of these marks remain stably fixed as a memory of previous stages (Ji *et al.*, 2010; Bock *et al.*, 2012; Farlik *et al.*, 2016). Of note, cells of the lymphoid lineage generally exhibit higher methylation than their myeloid counterparts, with promoters of myeloid TFs and their binding sites being frequently methylated (Bock *et al.*, 2012). The high specificity of methylation patterns enables the identification of predictive signature regions that can be used to reconstruct the hematopoietic tree (Farlik *et al.*, 2016). This knowledge has been exploited to identify the cell of origin in colorectal cancer (Bormann *et al.*, 2018) and several other cancer types (Zhu *et al.*, 2022). For these reasons, we believe that the similarity between CIMP and T-ALL samples is indicative that these leukemias derive from a cell that has been primed towards the lymphoid lineage. The proximity to the myeloid lineage that can be seen in both ATAC-seq and H3K27ac ChIP-seq data may be due to the multilineage potential retained by such an early progenitor.
- c) Mutational profile: although we did not identify any subgroup-defining mutations in CIMP leukemias, they exhibited frequent alterations in *NOTCH1*, *PHF6* and *JAK3*, all of which are recurrently mutated in T-ALL and rarely in AML. Other frequent genetic lesions were found in *WT1* and *ASXL1*, genes that can be mutated in AML, but also in ETP-ALL, especially the former. A systematic comparison of mutational profiles in CIMP, AML and T-ALL is available in Table S7 and has been described in more detail in the Supplementary Results. The overarching conclusion is that significant differences exist between mutational frequencies in CIMP and AML, but are smaller when comparing CIMP to T-ALL, and particularly to ETP-ALL. Based on this observation, it stands to reason that the cell of origin of CIMP is committed to the lymphoid lineage, in which mutations in genes like *NOTCH1* and *PHF6* preferentially initiate or maintain leukemogenesis.

In short, the lines of evidence presented here converge on an early hematopoietic progenitor as the cell of origin of CIMP leukemia. Transcriptional signatures only weakly suggest that this progenitor may be primed towards the lymphoid lineage, but this notion is further supported by clustering and mutational profiles. Although it remains somewhat speculative, a lymphoid-primed multipotent progenitor (LMPP) is compatible with these results. However, given the heterogeneity of these leukemias, it is entirely possible that this lymphoid bias is stronger in

some patients than in others. As mentioned above, we have modified the manuscript to clarify how we reach this conclusion. We have also tried to make it clear that the identity of this cell of origin is ultimately uncertain and may vary depending on the patient, even though it is almost certainly always a cell endowed with some degree of multipotency.

2: It is difficult to interpret that data in figure 3A as not statistical analysis is presented.

We have amended the figure with a statistical analysis as suggested by the reviewer, comparing the methylation score in CIMP against that of other leukemias and CD34+ cells. The reason we did not originally include p-values in the plot was that the number of data points for each genomic feature displayed in Figure 3A was extremely large. In null hypothesis significance testing, big sample sizes can result in very low p-values even when the differences between groups are trivial (Gómez-de-Mariscal *et al.*, 2021). Statistical tests attempt to ascertain whether the differences observed between groups are due to random sampling error, but this type of error decreases as the sample size approximates the population at large. At that point, even small differences that are biologically irrelevant may be sufficient to reject the null hypothesis, i.e., that there is no difference between the groups (Halsey, 2019). A preliminary analysis with the non-parametric Wilcoxon rank-sum test confirmed that the p-values were indeed minuscule for every comparison, certainly below the standard threshold of 0.05. Therefore, we reasoned that including them would be uninformative, or even potentially misleading. Instead, we chose to show only the descriptive statistics included in the boxplot to let the reader judge the magnitude of the difference.

That said, we now realize this was insufficient, especially given the large variance that complicates the interpretation of the data at first glance, as pointed out by the reviewer. Thus, we have conducted a more rigorous statistical analysis. First, we have used a Welch’s t-test instead of a non-parametric test. Even though the data do not follow a normal distribution, this test is in fact quite robust to deviations from this distribution when sample sizes are very large, by virtue of the central limit theorem (Fagerland, 2012). The Wilcoxon rank-sum test is useful for small sample sizes, but it becomes excessively sensitive to distribution differences in such large studies (Fagerland, 2012). Second, we have computed the effect size for each comparison using Cohen’s D, which standardizes the difference between two means by the pooled standard deviation. The result is the number of standard deviation units that separate two groups, with D=1 indicating that the mean of one group is 1 standard deviation larger than that of the other. Typically, values below 0.2 are considered small and above 0.8 are considered large (Sawilowsky, 2009). In our study, Cohen’s D was the highest when comparing promoters between CIMP and CD34+ cells (D=1.04) or AML (D=0.99), but much more modest for enhancers and gene bodies.

Since the p-values remain very small (in fact, so small that they are approximated to 0 because they reach the limits of machine precision), we have only included the effect size as measured by Cohen’s D in the revised Figure 3A. We also provide the table with the results of each comparison below:

Annotation	CIMP vs CD34	CIMP vs T-ALL	CIMP vs AML	Statistic
Enhancers	3.96E-137	4.60E-110	5.59E-304	p-value
	0.281	0.188	0.279	Cohen’s D
Promoters	0.000	0.000	0.000	p-value
	1.040	0.469	0.991	Cohen’s D
CTCF sites	0.000	0.000	0.000	p-value
	0.491	0.381	0.497	Cohen’s D
CpG	0.000	0.000	0.000	p-value
CpG	0.378	0.454	0.577	Cohen’s D
TSSs	0.000	0.000	0.000	p-value
TSSs	0.906	0.517	0.927	Cohen’s D

Gene bodies	0.000	0.000	0.000	p-value
Gene bodies	0.175	0.301	0.176	Cohen's D

3: The authors suggest that the silencing of transcription factors may be the cause of mixed phenotype as well as the differentiation block. While the latter is easily understood, it is not as easy to understand how silencing of activating transcription factors would result in aberrant gene expression. The discussion could be developed to address this subject in more detail.

We thank the reviewer for this insightful remark. Indeed, we suggest that silencing of TFs by promoter hypermethylation is a key driver of the aberrant transcriptional program of these leukemias, including both their differentiation arrest and their mixed phenotype. However, this is only one of several consequences of epigenetic dysregulation in these leukemias, which notably exhibit DNA hypermethylation, but also alterations in their enhancer landscape and chromatin organization. It should be noted that not all lineage-specific TFs are silenced in CIMP leukemias, and in fact some of them remain highly expressed compared to CD34+ cells or other leukemias, such as LEF1 or GATA3. There are multiple mechanisms whereby these epigenetic alterations can impact on gene expression beyond transcriptional repression, leading to an intermediate phenotype between the myeloid and lymphoid lineages. We discuss them below:

1. Silencing of repressive TFs: the existence of repressive TFs has been known since Jacob and Monod proposed that gene regulation was carried out by DNA-bound repressor molecules (Jacob and Monod, 1961). In fact, positive regulators were only discovered later, thanks to pioneering contributions of Ellis Englesberg on the L-arabinose operon (Englesberg *et al.*, 1965; Englesberg and Wilcox, 1974). More recently, a survey of human TFs identified 269 activators, 210 repressors and 115 bifunctional TFs based on their effector domains (Soto *et al.*, 2021). Studies of development in *Drosophila* have made it clear that a balance between both activators and repressors is required (Zeitlinger, 2020).

Both activation and repression play essential roles in hematopoietic differentiation, often mediated by TFs with bifunctional properties. In fact, a large number of the so-called “master regulators” simultaneously favor differentiation into one lineage while opposing the actions of TFs associated with alternative cell fates (Orkin and Zon, 2008). For example, PU.1 induces the activation of myelo-monocytic genes while suppressing the expression of erythroid/megakaryocytic TFs like GATA-1 (Nerlov and Graf, 1998). In turn, GATA-1 performs the opposite function (Kulesa, Frampton and Graf, 1995), in such a way that the balance between these two factors ultimately resolves the choice between lineages at the GMP stage (Rhodes *et al.*, 2005). Mutually antagonistic relationships like these operate at most key bifurcations in the hematopoietic hierarchy (Orkin and Zon, 2008). Shifts in expression of cross-antagonistic TFs dictate fate choice, to the extent that their enforced expression can reprogram even committed cells. Indeed, T-cell progenitors can be respecified into the myeloid lineage by C/EBP proteins and PU.1, but these effects can be reversed by NOTCH1 and GATA-3 (Laiosa *et al.*, 2006).

In view of the above, it is likely that some of the TFs silenced in CIMP leukemias act as repressors of alternative lineages, and therefore genes belonging to these programs become upregulated in their absence. As also noted in the discussion (page 9, lines 369-370), members of the C/EBP family favor commitment towards the myeloid lineage at the expense of lymphocytic differentiation (Laiosa *et al.*, 2006; Hasemann *et al.*, 2014). The loss of expression of these proteins (namely α , β and δ) observed in CIMPs could thus enable the continued expression of genes associated with the lymphoid lineage, in addition to blocking myeloid differentiation. Interestingly, C/EBP γ is upregulated in these leukemias, but it acts as a repressor of other C/EBP proteins. In fact, in a previous publication we demonstrated that downregulation of C/EBP γ is sufficient to restore neutrophilic differentiation. Similarly, we also point out in the discussion that “deletion of *LEF1* results in the upregulation of non-T-lymphoid genes via genome reorganization (Shan *et al.*, 2021), which could contribute to the mixed phenotype observed here”.

2. Preferential binding of TFs to methylated regions: in our study, we describe two ways, related but not identical, in which DNA methylation interferes with the activity of TFs. On the one hand, hypermethylation of CpG islands at promoters represses the expression of genes encoding for TFs. On the other, the binding sites of these TFs also become methylated, possibly preventing their binding at regulatory regions even if they are still expressed. The combination of the two effectively suppresses the action of TFs like C/EBP α and C/EBP β , both of which are downregulated and gain methylation at their binding sites. However, some TFs can specifically recognize methylated motifs, such as SP1 (Harrington *et al.*, 1988) or YY1 (Gaston and Fried, 1995). In fact, an unbiased screen determined that the binding of 23% of TFs is inhibited by methylation, but it was enhanced in 34% (Yin *et al.*, 2017). Therefore, it stands to reason that some TFs that exhibit preference for methylated CpGs bind novel regions that were previously inaccessible to them in CIMP leukemias. Others may simply not be affected by the methylation status of the cytosine. To investigate this further, we cross-referenced TFs that remain highly expressed in CIMP leukemias (TMM-normalized counts > 100) with a list of TFs that display preference for methylated sites in (Yin *et al.*, 2017). We identified several regulators of hematopoiesis, such as GATA3, NFATC3 or IRF2, which high affinity for methylated cytosines. This is largely speculative, but these are only some examples of TFs that remain abundantly expressed and can bind methylated regions, possibly leading to gene activation in CIMP leukemias.
3. Persistence of cell of origin programs: as discussed in the answer to comment 1.2, several lines of evidence point to an early hematopoietic progenitor as the cell of origin of CIMP leukemias, possibly with a bias towards the lymphoid lineage. Thus, the simultaneous expression of myeloid and lymphoid markers may be partially ascribed to the multilineage priming of these early progenitors (Hu *et al.*, 1997). The absence of TFs that antagonize opposite lineages enables the continued expression of these markers, which would be otherwise repressed upon cell commitment. At the same time, these cells still express high levels of genes involved in HSC maintenance and self-renewal, such as *ETV6* (Hock *et al.*, 2004), *LYL1* (Souroullas *et al.*, 2009) or *ELF1*, among others (Xiang *et al.*, 2010) (Table S9). In sum, the differentiation block caused by the silencing of TFs critically involved in cell commitment may preserve a pre-existent hybrid transcriptional program that would normally be transiently present in early progenitors.
4. Region-specific demethylation: although CIMP leukemias are defined by extensive DNA hypermethylation relative to other leukemias, large swaths of their genome remain in an unmethylated state. As shown in Figure 4, some promoters exhibit less methylation than in AML and, especially, T-ALL. Few of those are key hematopoietic TFs (labeled in red in the plot), but they may nonetheless play a role in gene regulation. More importantly is the large number of genes, depicted in grey in the image, whose expression and methylation remain unaltered with respect to AML or T-ALL. These include TFs that are expressed at similar levels as in myeloid or lymphoid cell types, thus enabling a certain degree of lineage commitment, which is ultimately interrupted by the absence of essential master regulators.
5. Chromatin rewiring: given the extensive DNA hypermethylation in these leukemias, we reasoned that the binding of CTCF, which is known to have less affinity for methylated cytosines, would be disrupted. In page 7 we showed that this is indeed the case with ChIP-seq data, particularly in regions with CpG dinucleotides within the CTCF motif (Figure 6). Even though we expected this would lead to major alterations in 3D genome topology, analysis of Hi-C data revealed only a few changes in TAD boundaries and chromatin loops (Figure 7). Some of these changes, however, were enough to induce expression of hematopoietic TFs like *GATA3* or repress others, including *CEBPD* and *KLF4*. Furthermore, some dysregulated TFs are involved in chromatin organization independently of CTCF -- we mention the examples of *LEF1* (Shan *et al.*, 2021) and *KLF4* (Wei *et al.*, 2013; Di Giammartino *et al.*, 2019) in the discussion. Inasmuch as our analysis is limited by the resolution and depth of our Hi-C data, it is possible that perturbations in other loops have gone unnoticed. In any case,

these few examples illustrate how genome-wide DNA methylation may result in overexpression of certain key genes.

6. High expression of some lineage-specific TFs: despite the silencing of many lineage-associated TFs, including CEBP proteins or KLF4 on the myeloid side, and LEF1 on the lymphoid side, others remain expressed at sufficient levels to exert trans-regulatory effects. For example, lymphoid TFs such as BCL11B (Longabaugh *et al.*, 2017) or GATA3 are expressed at relatively high levels compared to AML or CD34+ cells. Although PU.1 (*SPI1*) is less expressed in CIMP than in AML or in CD34+ cells, its binding sites are significantly more accessible than in T-ALL (Figure 5B), which possibly maintains the myeloid phenotype of these leukemias. The intermediate expression/activity of these TFs (lower than in their corresponding lineage, but certainly higher than in the alternative trajectory) is consistent with the mixed phenotype of these leukemias. Different causes may underlie each of these events, as described in this answer: lymphoid TFs may become more expressed in the absence of C/EBP-mediated repressor, PU.1 activity may be a vestige of the cell of origin, *GATA3* expression stems from the formation of novel loops, etc.

In summary, DNA methylation is a very likely driver of leukemogenesis in these leukemias, but the downstream consequences of this event are complex and extend beyond mere transcriptional repression. Promoter hypermethylation silences the expression of both activators and repressors, arresting differentiation at early stage of hematopoiesis. This enables the continued expression of genes typically associated with alternative lineages, many of which may be already expressed in the cell of origin. Indeed, ATAC-seq and H3K27ac ChIP-seq data reveal that myeloid enhancers are frequently active in CIMP leukemias. Furthermore, methylation of regulatory regions interferes with the binding of some TFs but favors others, whose preferential binding to newly methylated sites can stimulate the expression of nearby genes. Among the proteins whose binding is altered is CTCF, whose absence perturbs chromatin organization. Upon formation of novel loops and disruption of TAD boundaries, genes may also become aberrantly expressed.

We have incorporated parts of the text presented above to the Discussion section of the manuscript to better explain how these different elements of gene regulation converge on the aberrant phenotype of CIMP leukemias. However, given the manuscript was already rather long in its original version, we tried to keep this additional explanation concise.

Reviewer #2, expertise in leukemia genomics and epigenomics (Remarks to the Author):

The work by Mulet-Lazaro et al. and colleagues uses orthogonal functional genomics assays to characterize the genomic, epigenomic and transcriptomic landscape of mixed myeloid/lymphoid leukemia. The authors identified that epigenetic dysregulation, rather than genetic alterations, drive mixed lineage leukemia. Specifically, mixed lineage leukemias appear to harbor a hybrid epigenetic landscape, with methylation patterns consistent with T-ALL, but a regulatory landscape (H3K27ac and ATAC-seq) more consistent with AML. Overall, this manuscript provided interesting insights on mixed lineage leukemias. One prominent drawback was the descriptive nature of the work, with no functional validation or mechanistic insights beyond genomic annotation. Key points the authors should address include:

We acknowledge that the lack of experimental validation was, indeed, a limitation in our work. Following the suggestions of this reviewer in comments 2.6 and 2.7, we have incorporated some functional work that, we hope, will address these concerns.

1. In Figure 1A, is it possible to add T-ALL data to show closer relationship of CIMP leukemias with T-ALL versus AML? This is shown with UMAP analysis in Figure 1C but would also be relevant for Figure 1A.

We thank the reviewer for the suggestion, but we would like to point out that Figure S2D, present in the original version of the manuscript, showed the relationship between different leukemias as a Pearson correlation heatmap. We referred to said figure at the end of page 3 (lines 122-123 of the revised version).

The reason the first heatmap presented in Figure 1A did not include T-ALL cases was that CIMP leukemias were originally diagnosed as AMLs on the basis of their immunophenotypic markers. The main purpose of that figure was to show that, based on their methylation profile, a) CIMPs were markedly different from the majority of AMLs, b) cases identified by the German and Dutch groups belonged to the same disease entity. Indeed, all CIMP cases (known as CEBPA-silenced in a previous publication) clustered together, and apart from all AMLs. The only exception was a case with double *CEBPA* mutations in the C-terminal domain.

As pointed out by the reviewer, CIMP cases are indeed more closely related to the majority of T-ALL cases included in our study, at least based on DNA methylation. This is apparent both in the UMAP (Figure 1C) and in the heatmap (Figure S2D), in which the following T-ALL cases cluster with CIMP leukemias: #T174, #T059, #T069, #T145, #T162, #UKRT3. There were two cases that were only grouped together with CIMPs in the UMAP (#UKRT2, #UKRT1), possibly due to differences in the inner workings of each algorithm. In any case, none of the AML cases clustered with CIMP leukemias either in the UMAP or in Pearson correlation heatmap.

2. In Figure 1C, several T-ALL samples cluster with AML for MCIP-seq, and several cluster with both AML and CIMP in the H3K27ac and ATAC-seq analyses. Are these ETP-ALL samples? Are they the same samples across the panels? The authors should clarify this. If these are indeed ETP-ALL samples, they should be distinguished from the remaining T-ALL samples on the figure panels.

Unfortunately, the T-ALL samples analyzed with for MCIP-seq were not available when we generated the other data employed in this study, so the patients in that panel are inevitably different from those in the ATAC-seq and H3K27ac ChIP-seq analyses. We indicated in Table S1 what datasets were available for each patient, as well as their corresponding phenotype, but we realize it should be clearer in the manuscript. We have added a note in this respect in the caption of Figure 1 (page 13, line 503), as well as in the Methods section (page 10, line 443).

Since MCIP-seq was generated in a different set of T-ALL patients, we examined this dataset separately from the other two. Out of 14 T-ALL samples, 8 clustered with CIMPs in the UMAP, one with AML (#T034) and 5 formed an independent cluster (Figure R1). We classified T-ALL cases into ETP or non-ETP in accordance with the original definition in (Coustan-Smith *et al.*, 2009), later incorporated into the World Health Organization (WHO) classification (Arber *et al.*, 2016). Thus, we identified as ETP-ALL samples with blasts that expressed CD7, lacked CD1a and CD8, and were positive for one or more of the myeloid or stem cell markers CD34, CD117/KIT, HLADR, CD13, CD33, CD11b, or CD65. Immunophenotypic information was available for 9 out of 14 T-ALL cases included in the MCIP-seq experiment, but only the patient that clustered with AMLs was classified as ETP-ALL (#T034). All the other cases, including those that exhibited similarity with CIMPs in the UMAP, were non-ETP with total absence of myeloid markers and clear expression of lymphoid markers.

Figure R1. Uniform Manifold Approximation and Projection (UMAP) of MCIP-seq data.

In the ATAC-seq and the H3K27ac ChIP-seq analyses, we could confirm overlap between the T-ALL samples that clustered together with CIMPs and AMLs (Figure R2). More specifically, all 3 T-ALL cases that clustered with CIMP and AML in the H3K27ac UMAP (#13547, #14853 and #14855) did so in the ATAC-seq data as well. These also

clustered similarly in the corresponding Pearson correlation heatmaps (Figures S2F, S2G). Nevertheless, one patient (#14969) was only grouped with AML in the ATAC-seq UMAP, but not in the H3K27ac one. In the RNA-seq data, none of these patients clustered close to the majority of CIMP cases, albeit #13547 colocalized with a few CIMPs that seemed to have a more lymphoid transcriptional program.

Figure R2. UMAP of H3K27ac ChIP-seq, ATAC-seq and RNA-seq data. Cases that cluster together with AMLs in both the ATAC-seq and H3K27ac data are highlighted in red.

To find out if any of these 4 borderline cases were ETP-ALL, we next classified them based on their immunophenotype as described above. Contrary to our expectations, two of these samples (#14853 and #14855) were not classified as ETP-ALL, whereas the other did not have immunophenotypic information available. Interestingly, #14969, which only clustered with AMLs and CIMPs in the ATAC-seq analyses, was classified as ETP-ALL. That patient also carried *NOTCH1*, *PHF6* and *SUZ12* mutations, all of which were very common among CIMPs. Following their identification based on immunophenotypic data, we labeled all the ETP-ALL cases in UMAPs presented here, as well as in the revised version of Figure S2. This additional information makes it possible to assess how ETP-ALL relates to CIMP and other leukemias in a broader context, albeit the sample size is too small to obtain a definitive answer. As discussed above, ETP-ALL cases cluster together with most other T-ALLs in the H3K27ac and ATAC-seq (Figure S2C, center and right panels). In terms of gene expression, ETP-ALL and CIMP appear as two entities of the continuum of transcriptional programs ranging from AML to T-ALL, with ETP-ALL having a slightly more lymphoid phenotype (Figure S2C, left panel). This is further confirmed by the PCA plot, in which CIMP is more proximal to the myeloid leukemias than ETP-ALL is.

In conclusion, clustering analyses reveal differences between CIMP and ETP-ALL at the epigenetic level, indicating they are biologically distinct despite similarities in transcriptional signatures, mutational profiles and cell of origin. CIMP leukemias harbor a regulatory landscape that retains significant myeloid potential, which is nevertheless forestalled by aberrant methylation. In contrast, the cis-regulatory element (CRE) repertoire in ETP-ALL resembles that of other T-ALLs, with notable differences from CIMP at the enhancers of master regulators like *CEBPA*. We have integrated these additional findings into the revised manuscript by incorporating new figures labeled with the ETP status in the Supplement and making the corresponding modifications in the Results (page 4, line 119) and Discussion (page 8, lines 330-333) sections.

3. Because the authors perform ATAC-seq, an analysis of TF footprints should be performed to identify candidate bound TFs at sites of interest. Moreover, comparisons of TF footprint intensity between CIMP vs AML or T-ALL can further identify key TFs that show differential activity. These comparisons can be performed at a subset of sites of interest using computational tools such as TOBIAS.

We thank the reviewer for this suggestion. Indeed, the availability of ATAC-seq data enables the estimation of TF accessibility based on footprints. Nevertheless, such analyses generally require a large depth of coverage, with an usual recommendation being 200 million reads for optimal results (Yan *et al.*, 2020). A typical approach to overcome this limitation is to merge biological replicates to achieve the recommended depth; in fact, TOBIAS lacks support for replicates and explicitly recommends users to merge them. While this is useful for the identification of consistent differences across replicates, an important caveat is that it masks the heterogeneity of the diseases under study. Therefore, we decided to use chromVAR instead, which calculates, for every sample, deviations of chromatin accessibility between the number of reads mapping to peaks containing a motif and the expected number based on the average of all samples (Schep *et al.*, 2017). Using this method, which effectively preserves replicate information, we identified C/EBP, SPI1, RUNX and FOXO TFs, among others, as key drivers of transcriptional differences between CIMP and AML or T-ALL (Figure 5). We could also measure the correlation between motif accessibility and the transcription of the corresponding TFs, which allowed us to pinpoint key activators that become activated or silenced in CIMP.

That said, a downside of chromVAR is that it does not take the actual TF footprint into account, assuming instead that a motif contained within an accessible region is bound. Furthermore, it cannot identify candidate TFs bound at specific regions of interest. In view of these limitations, we have now conducted a footprinting analysis with TOBIAS, as suggested by the reviewer. Like in the previous analysis, we employed human motifs from the 2022 version of JASPAR, filtering out those corresponding to genes not expressed in any leukemic or CD34+ samples. Reassuringly, the comparisons between CIMP and other leukemias yielded very similar results to the chromVAR analysis, including loss of binding of C/EBP proteins relative to AML or gain of SPI1 activity relative to T-ALL. Moreover, we used the footprints computed by TOBIAS to identify TFs that lose binding to methylated regions, such as the promoters of *CEBPA* and *KLF4*.

The results of this analysis have been included in Figures S8C-E and described in the corresponding segment of the Results section (page 6, lines 235-241).

4. In Figure 2A, if the authors performed clustering analysis of methylation and epigenomic signatures in just the CIMP leukemias, is any genetic lesion(s) shared/common among samples that cluster together?

This is an intriguing possibility, but it should be noted that such analyses are unfortunately hindered by the relatively small sample size at our disposal. Even so, we attempted to identify possible subgroups in CIMP leukemias and mutations associated with them, as suggested by the reviewer. To that end, we visualized the Pearson correlation between samples as a hierarchically clustered heatmap and overlaid mutations recurrent in these leukemias. We selected genes found mutated at least in 25% of the cases, plus others selected because they impact on genes of potential interest (*TP53*, *DNMT3A*, *IDH2* and *LEF1*).

The analysis of MCIP-seq data revealed a small cluster of 3 patients highly correlated with each other that shared mutations in both *DNMT3A* and *IDH2*, absent in all other cases (Figure R3, left). This is particularly noteworthy given the role of the enzymes encoded by these two genes in DNA methylation and the hypermethylation status of CIMP leukemias. It has been previously shown that mutations in both *DNMT3A* and *IDH2* or its homolog *IDH1* define a subgroup of AML by hierarchical clustering of enhanced reduced representation bisulfite sequencing (ERRBS) (Glass *et al.*, 2017). Thus, it is possible that these mutations confer unique methylation patterns to these 3 samples as well. Nevertheless, it is unlikely they are responsible for their global hypermethylation. Several AML cases in our cohort, namely #AML107, #AML0100 and #AML0132, carried these mutations but clustered firmly together with other AMLs (Figure 1A). It is also worth mentioning that the 3 *DNMT3A/IDH2*-mutated cases carried fewer mutations (Figure 2A) and gross chromosomal aberrations (Figure 2B) than most other cases. Furthermore, two of these *DNMT3A/IDH2*-mutated patients equally clustered together in the H3K27ac and ATAC-seq data, with the remaining one being missing in these datasets (Figure R3).

Figure R3. Pearson correlation heatmaps with dendrograms indicating hierarchical clustering. Two subgroups were independently defined for each dataset by automatically locating the cutting point in the dendrogram that yielded the desired number of clusters. The column names in the heatmap correspond to patient identifiers, whereas the annotation rows at the top indicate the mutational status of each selected gene (yellow = wild type, red = mutation). PCC = Pearson correlation coefficient.

Interestingly, a subgroup identified in the H3K27ac CHIP-seq data (albeit with only $PCC \sim 0.3$) exhibited *FBXW7* mutations. These are known to disrupt the degradation of the NOTCH1 intracellular domain (NICD) normally mediated by this protein, thereby resulting in sustained activation of the NOTCH pathway (O’Neil *et al.*, 2007; Thompson *et al.*, 2007; Close *et al.*, 2019). This is an alternative mechanism to the mutations in the PEST domain that result in constitutive activity of NOTCH1, frequently seen in T-ALL (Weng *et al.*, 2004) and chronic lymphocytic leukemia (CLL) (Filarsky *et al.*, 2016), but also in almost 50% of CIMP leukemias. However, patients with such *NOTCH1* mutations exhibited weaker correlations with *FBXW7*-mutated cases, seemingly excluding the possibility that activation of NOTCH pathway is responsible for their distinct epigenetic profile. Furthermore, only two of these cases clustered together in the ATAC-seq heatmap, with the third exhibiting significantly less correlation.

In conclusion, integrative analysis of epigenetic-driven clustering and mutational profiles revealed a few mutations that may define CIMP subgroups, but their biological significance is unclear. In any case, the sample size was too small to conclusively prove that these commonalities did not simply arise by chance.

5. For the transcriptional analysis, the authors should try to integrate these data with their epigenomic profiling. For examples, do the authors identify any correlation in epigenetic changes at regulatory sites (H3K27ac/ATAC) between CIMP vs ETP-ALL or CIMP vs AML and the corresponding transcriptional effects on associated genes between CIMP vs ETP-ALL or CIMP vs AML? For instance, does reduced/enhanced H3K27ac signal in CIMP vs AML lead to reduced/enhanced transcription of a neighboring gene in CIMP vs AML? Of course, assigning a target gene to regulatory elements can be difficult and prone to error. However, I would expect some level of correlation to further demonstrate epigenetic changes in CIMP have functional consequences on transcription.

We thank the reviewer for this excellent suggestion. Indeed, epigenetic changes are only relevant insofar as they have an impact on the transcriptional program of the cells. In our original manuscript, we showed this to be the case for DNA methylation, which is arguably the most notable feature of these leukemias, but indeed such an analysis is lacking for other epigenetic marks. We did not initially examine the relationship between H3K27ac/ATAC-seq and gene expression due to length considerations. Nevertheless, we have now performed this additional analysis to give a more comprehensive picture of the interplay between epigenome and transcriptome in these leukemias.

Identification of the genes targeted by regulatory elements is certainly a challenging task. The simplest approach is to link every regulatory element to the closest promoter, but this strategy only identifies the correct gene in about 40% of the cases. (Sanyal *et al.*, 2012; Andersson *et al.*, 2014; Fulco *et al.*, 2019; Gasperini *et al.*, 2019). Another frequently employed strategy relies on the correlation between features of promoters and enhancers, such as open chromatin or H3K27ac (Ernst *et al.*, 2011). This is often used in the analysis of single cell ATAC-seq with tools such as Cicero (Pliner *et al.*, 2018). However, an important caveat is that these correlations may be affected by additional regulatory processes that interfere with gene expression. For example, the +42 kb enhancer of *CEBPA*, whose importance in hematopoiesis has been well-established *in vivo* (Avellino *et al.*, 2016), exhibited Pearson correlation coefficient of 0.54 in the ATAC-seq data of our cohort. As shown in Figure R4, this is due to the lack of correlation for most CIMP samples, in which the gene is silenced by DNA hypermethylation even though the enhancer remains active.

In view of these limitations, we integrated multiple layers of information in order to increase the accuracy of the CRE-promoter assignment, namely: distance to nearest promoter, correlation between ATAC-seq peaks, co-occurrence of both elements in the same TAD, contact by a loop, previous assignment by the GeneHancer database (Fishilevich *et al.*, 2017). As an example of the validity of this approach, an enhancer that would be assigned to *RPN1* based on proximity alone was correctly linked to *GATA2*.

Figure R4. Scatterplots displaying the correlation between ATAC-seq signal at the *CEBPA* promoter and various putative enhancers in a 50 kb window around the TSS.

Next, we used *DiffBind* to compute differential ATAC-seq signal between putative enhancers (defined as peaks not overlapping with a TSS \pm 500 bp) successfully assigned to a target gene. For each of these genes, differences in the accessibility of their enhancers were linked to differences in their expression. The results of this analysis were included in Table S24 and Figure 4C, and summarized in the corresponding Results section (page 6, lines 200-209). We performed an equivalent analysis with H3K27ac ChIP-seq data instead of ATAC-seq, presented in Table S25 and Figure S6C, which yielded comparable results. As predicted by the reviewer, there is correlation ($\rho \sim 0.18$) between changes in accessibility and expression when comparing CIMP against either AML or T-ALL. Although this value is somewhat modest, it should be noted that it is the aggregate of all genome-wide changes across multiple patients. Furthermore, as discussed before, CIMP leukemias retain active enhancers of myeloid genes that are silenced by DNA methylation, resulting in accessibility patterns that do not always match transcriptional status.

6. The authors should perform ChIP-seq for some key TFs that they claim are altered in CIMP versus AML and T-ALL samples (e.g., *CEBPA*, *KLF4*, *RUNX1*, etc.). This can also be performed in T-ALL or AML cell lines and the sites compared to the corresponding sites in patient samples exhibiting altered epigenetic state in CIMP (or reduced TF footprinting; see comment 3 above). Some of these TF profiles may already be available from ENCODE in related cell types.

We agree with the reviewer that it would be ideal to perform ChIP-seq for key TFs in primary cells of leukemia patients. Unfortunately, said material was scarce after having performed the numerous high throughput experiments included in our study. Thus, it was not feasible to evaluate the binding of the wide range of TFs

whose expression and/or binding are dysregulated in CIMP leukemias, according to our previous analyses. In the original version of the manuscript, we attempted to circumvent this limitation by calculating the significance of the overlap between differentially methylated regions (DMRs) and publicly available TF binding data. To this end, we employed the *LOLA R* package, which includes data on hematopoietic cell lines, leukemia patients and healthy hematopoiesis from ENCODE and CODEX. The results showed that regions hypermethylated in CIMP relative to AML were enriched for binding sites of TFs like CEBP/β, PU.1 and RUNX1, among others; surprisingly, CEBP/β and PU.1 were also overrepresented in hypomethylated regions (Figures 5A and S7A-D). When comparing CIMP against T-ALL, hypermethylated regions were enriched for the lymphoid TFs such as MYB and NOTCH1, among others, whereas binding sites for myelo-erythroid TFs (SPI1, CEBP/α and β, GATA1) were often present in demethylated areas. These observations, together with other evidence gathered in the study, led us to conclude that hypermethylation of regulatory regions in CIMP leukemias plays an important role in restricting their ability to differentiate into either the lymphoid or the myeloid lineage. However, the lack of methylation at many positions typically occupied by myeloid TFs seems to indicate that they retain a significant myeloid potential. In our view, this analysis is similar to the one proposed by the reviewer as an alternative to ChIP-seq in CIMP patients – we indeed examined TF profiles derived from other leukemia patients or cell lines in sites exhibiting an altered epigenetic state in CIMP patients.

Nevertheless, we realize this approach is an imperfect substitute that cannot truly replace actual experimental data in the patients themselves. Therefore, we complemented the previously mentioned analyses with ChIP-seq for TFs that are particularly relevant in this disease context: CEBPA, SPI1 (PU.1) and TCF7. In all these cases, the ChIP-seq results were in agreement with the predictions derived from open chromatin and DNA methylation data: CEBPA binding was practically absent in CIMP leukemias and SPI1 was increased relative to T-ALL (but decreased compared to AML), whereas the reverse was true for TCF7. We have included several figures depicting both genome-wide and locus-specific changes in TF binding in Figure S9. These results are also described in page 6, lines 241-246.

7. Can the authors functionally demonstrate that their work provides therapeutic insights as CIMP leukemias overall have a poor prognosis? For instance, as the authors suggest, would NOTCH1-targeting drugs or decitabine be a viable treatment in CIMP PDXs? Are CIMP PDXs/patient samples more sensitive to NOTCH1-targeting drugs/decitabine compared to AML PDXs/samples? This experimentation would provide translational value and validation to an otherwise descriptive manuscript.

In order to verify whether the proposed therapeutic options are effective against CIMP leukemias, one could study their effect in primary cell cultures or in PDX models. Given that material is limited for these rare cases and that we would have to test for each CIMP whether it can grow *in vivo*, we decided to set up experiments in primary cell cultures. As a demethylating agent, we chose 5-azacitidine, a pyrimidine nucleoside analogue widely used in clinical care that incorporates into DNA and covalently traps DNA methyltransferases, which are subsequently degraded by DNA damage response (Stresemann and Lyko, 2008). Independently of this demethylating effect, 5-azacitidine can also induce cytotoxicity in an unspecific manner. To target the NOTCH pathway we used nirogacestat, a gamma secretase inhibitor recently approved by the FDA that prevents the cleavage of the NOTCH intracellular domain and its subsequent translocation to the nucleus (López-Guerra *et al.*, 2015). We first performed dose-response experiments in cell lines¹ to select a suitable dose of each compound, using the existing literature as a guidance (data not shown).

Then, we tested the compounds in patient cells from CIMP, AML and T-ALL for 7 days and measured the number of cells at days 2, 4 and 7 (Figure R5A). Of note, CIMP cells started dying out after day 2 even in the absence of any treatment, and there were practically gone by day 7. Thus, it would have not been possible to continue the

¹ T-ALL cell lines: ALL-SIL, Loucy, DND-41, Jurkat, MOLT-4. AML cell lines: HL-60, U-937.

experiment beyond that time point. We cultured AML and CIMP cells with a combination of myeloid factors (SCF/KITL, FLT3L, TPO, IL6 and IL3), whereas T-ALL cells were grown with a cocktail that stimulates lymphoid cells (SCF/KITL, FLT3L and IL7). All leukemias responded to the treatment with 5-azacitidine (Figure R5B). Despite the higher methylation levels of the CIMP cases, they were not uniquely responsive to 5-azacitidine, suggesting that we mainly observed its direct cytotoxic effects. Supporting this notion, the rapid decline of living cells in the DMSO-treated group indicates that CIMP blasts did not divide in culture. As a result, they did not incorporate 5-azacitidine into the DNA during replication, which is necessary to induce demethylation (Stresemann and Lyko, 2008).

There was no response to nirogacestat in any leukemias relative to controls, despite the fact that both the CIMP and T-ALL cases harbored *NOTCH1* mutations (Figure R5A). A likely explanation is that the experiments were too short to observe a therapeutic impact, in keeping with previous publications that only reported growth inhibition of treated cell lines after 6-7 days (Govaerts *et al.*, 2021). Unfortunately, by that time most CIMP cells were already dead in our culture, which precluded a longer experiment as mentioned above.

Since CIMP leukemias did not survive long in these initial experiments, we next investigated if it was possible to culture them longer with other media. Thus, we compared their survival in media with a) no growth factors, b) myeloid factors, or c) lymphoid factors. As expected, AML cells responded well to the myeloid cocktail, although they also survived in large numbers with the lymphoid medium, possibly due to the presence of FLT3L and SCF in this cocktail. T-ALLs only responded to lymphoid factor, whereas CIMP failed to respond to all tested cytokine combinations. This is also consistent with previous observations revealing that CIMP leukemias do not respond to IL3, GM-CSF or G-CSF (Figueroa *et al.*, 2009).

All in all, the limited survival of CIMP cells *in vitro* prevents the study of either demethylating agents or *NOTCH1* inhibition on these samples. Therefore, we are only able to speculate on the clinical efficacy of those compounds on leukemia development in those CIMP patients.

Figure R5. Results of primary cell culture experiments. A. Flow cytometry density dot plots of a representative CIMP, AML or T-ALL sample treated with DMSO, 5-azacitidine or nirogacestat for 7 days. The plots show forward scatter (FSC) on the X axis and side scatter on the Y axis (SSC), with the gating indicated by a black contour line and the number of living cells shown at the top right. **B.** Line plot showing the average (n=2) number of living cells following treatment of AML, CIMP or T-ALL with either 5-azacitidine (orange) or DMSO (grey). The bars depict the standard error of the mean. **C.** Flow cytometry density plots of CIMP, AML or T-ALL samples (n=2 each) cultured with media containing no growth factors, myeloid factors or lymphoid factors.

Reviewer #3, expertise in the classification of acute leukemias of ambiguous lineage, CIMP and epigenetics (Remarks to the Author):

The study by Mulet-Lazaro et al entitled title 'Epigenetic alterations affecting hematopoietic regulatory networks as drivers of mixed myeloid/lymphoid leukemia' used a multi-omic approach to evaluate the role of the epigenetic landscape in leukemias with ambiguous lineage. Of particular focus is a hypermethylated AML subtype which they labelled CIMP. This labeling is unfortunately confusing since CIMP subtyping is also used for subgrouping of e.g. T-cell leukemias (T-ALL CIMP negative and CIMP positive). In this manuscript, both the CIMP AML group (included in most figures) and the CIMP T-ALL subgroups are analyzed (Figure S2B). To avoid this confusion, a more specific naming of the hypermethylated AML subtype would have helped. The overall aim is relevant but unfortunately, the methods and the results (figures/tables) are not presented clearly in their current form.

We thank the reviewer for the insightful remarks. Although we understand how this labeling can be confusing, we would like to point out that the term CIMP has been previously used to define subgroups of other malignancies that exhibit CpG hypermethylation. It was coined in 1999 by the group of Jean Pierre Issa in the context of colorectal cancer (Toyota *et al.*, 1999), but it has been subsequently applied to bladder, breast, endometrial, gastric, ovarian or lung cancer, among many others (Hughes *et al.*, 2013). In AML, aberrant methylation of CpG islands suggestive of a hypermethylator phenotype was documented as early as 1999 (Melki, Vincent and Clark, 1999), and shortly afterwards by the Issa group (Toyota *et al.*, 2001). The same team also published similar findings in ALL just one year later (Garcia-Manero *et al.*, 2002). The term CIMP does not appear in these publications, but a review of 2004 by Dr. Issa indicates that these findings are “reminiscent of CIMP”. Other reviews have also considered these studies as evidence of a CIMP subgroup in both T-ALL and AML (Hughes *et al.*, 2013). In later studies, the label CIMP has been explicitly used to classify patient subgroups in both T-ALL (Roman-Gomez *et al.*, 2005; Borssén *et al.*, 2013; Touzart *et al.*, 2021) and AML (Kelly *et al.*, 2017; Gebhard *et al.*, 2019). Therefore, we believe our use of that term is justified and in line with the existent literature; as a matter of fact, 5 patients of our cohort had originally been classified as CIMP by Gebhard *et al.*

The remaining leukemias in our cohort had been referred to as “CEBPA silenced” in prior publications (Wouters *et al.*, 2007; Figueroa *et al.*, 2009; Alberich-Jordà *et al.*, 2012), owing to the fact that they were identified as a part of a gene expression cluster characterized by loss of either function or expression of *CEBPA*. We considered keeping that nomenclature, but in the course of the current study we realized that, while loss of *CEBPA* is likely a key oncogenic driver in these leukemias, they exhibit hypermethylation-driven silencing of many other TFs. Therefore, we decided to use the “CIMP” label instead because it captures the genome-wide nature of this phenomenon and had been previously used by some authors of this publication. Even though we acknowledge that the simultaneous use of CIMP for a subgroup of T-ALL in the analysis of MethylationArray data can be confusing, this is restricted to a single figure of the supplement. This should be clear in the corresponding legend of the figure, but we amended the corresponding Results section to make the reader aware of the distinction (page 3, line 117).

Finally, we would like to note that the leukemias in our cohort exhibit a mixed myeloid/lymphoid phenotype, albeit they were originally identified as a hypermethylated subgroup of AML. This is not only apparent at the level of surface markers, but also in terms of transcriptional programs and epigenetic control. Therefore, we have abstained from labeling them as “CIMP AML”, and instead opted for a more neutral “CIMP leukemias”.

The overall aim is relevant but unfortunately, the methods and the results (figures/tables) are not presented clearly in their current form.

We thank the reviewer for this critical remark, which has been addressed in more detail below. One of the reasons for this lack of clarity is that an additional Supplementary Information file, containing a detailed description of the methods and the legends of the Supplementary Figures, was not uploaded to the submission portal. Instead, only the Supplementary Figures were uploaded on their own. We sincerely apologize for this oversight, which has been amended in the revision. Other potentially confusing aspects such as the presence of apparently redundant figures can also be explained by additional analyses that are only presented in the Supplementary

Results. Moreover, we have made an effort to eliminate less relevant figures, based on the recommendations of the review in the point-by-point comments.

Major comments:

⌘ Even though the number of main figures is within the allowed number (10 figures/tables), the number of individual plots is extremely high (n=36 individual main figures and n=86 individual Supplementary figures, along with 38 supplementary tables). Some figures are shown both as main figures and supplementary figures (e.g. Fig 1C, Fig S2C). Not all figures and tables are referred to in the text and those one should be removed. There are results presented in the discussion section that have not been presented in the result section (Figure S12A-B), and they are recommended to be added to the result section.

We acknowledge that the number of individual plots is very high, but this is a consequence of the comprehensive nature of this study, which features a large number of datasets and analyses with the overall aim of fully characterizing these leukemias. That said, we have reduced the number of redundant or uninformative figures in the revised version. However, we also had to address the requests of other reviewers, resulting in the generation of additional plots. Therefore, the overall number of visual elements has not decreased dramatically in this new version, but we have tried to keep only those that were relevant.

To the best of our knowledge, the only figure that was duplicated (by mistake) was Figure S1D, which was the same as S2A. This has been amended in the current version. The UMAP plots in Figure S2C are not exactly the same as those presented in Figure 1C, since they contain additional information about leukemia subgroups. Although we could have used them as main figures, we chose to keep them separate to emphasize that the main message of that section was about the relationship between CIMP, AML and T-ALL. The presence of additional colors for each subgroup could be distracting and make these relationships less obvious to the reader.

As suggested, we have eliminated figures and tables not referred to in the manuscript. Nevertheless, many elements that may have seemed to be uninformative are mentioned in the Supplementary Results, which were unfortunately missing from the original submission. This is the case, for example, of Tables S6-8, summarizing recurrent mutations identified in other leukemia cohorts. The figures that we have removed because they were of little consequence to the study are the following, listed with their original label: Figure S3E-F (showing copy number alterations in AML and T-ALL), S4A (GSEA enrichment plot), S4D (heatmap with the results of *CIBERSORTx*, already presented in S4C), S8A and S8C (clustering on the basis of deviations calculated by *chromVar*), S10A (genomic annotation of CTCF ChIP-seq peaks), S10B (showing motif enrichment in CTCF peaks) and S10C (volcano plot for CTCF ChIP-seq). Furthermore, we added references to figures that we failed to mention in the previous version, but were relevant to the study: Figure S5G (page 5, line 187), Figure S5N (page 5, line 191), Figure S11A (page 7, line 267).

Regarding Figure S12A-B (S13 in the new version), we chose to present it in the Discussion because it did not belong in any specific subsection of the Results, but it was valuable when discussing the role of *CEBPA* in these leukemias and the differences between their methylome and their enhancer landscape. We have added a reference to this figure in a section of the Supplementary Results that elaborates on the dysregulation of C/EBP proteins in CIMP leukemias (page 5 of that document).

⌘ The method description in the main text is very short and do not declare all method used (e.g. not the array data), and the patient material needs to be described in more detail (age, numbers, etc.). An extended description of the material and methods is needed, and are currently not included as Supplementary Information as claimed in the manuscript (row 430-431). These method details are necessary to evaluate the results and conclusions of the manuscript.

We claimed that there was an extended description of the Methods in the Supplementary Information because that was indeed the case, but we failed to upload that document to the submission portal. Once again, we

apologize for this omission, which has been addressed in the current version. We hope you will find the Supplementary Methods sufficiently descriptive.

Furthermore, we described in more detail the characteristics of the patients included in our study as suggested by the reviewer (page 11, lines 439-449). A summary of the data generated is also available in Table S1.

⌘ The Supplementary Figures lack figure legends and cannot be interpreted without this information. Several shortenings in the figure axis are not explained in the figure or text.

The legends of the Supplementary Figures can be found in the Supplementary Information file that was missing from the original submission. They should be available in the current version.

⌘ A descriptive title describing what the tables show would have been helpful.

Likewise, the titles of the tables were included in Supplementary Information file that we omitted from the submission by mistake. We apologize once again for the inconvenience.

⌘ Several figures lack information on the number of samples included in each analysis, some lack statistics and a description of subgrouping definitions (e.g. mutation subgrouping) making it difficult to interpret the results and significance of the data.

We thank the reviewer for bringing this shortcoming to our attention. We have adjusted the figure legends, both in the main text and the supplement, to include detailed statistical information, such as the number of patients per group and the statistical tests employed. To that end, we have also relied on the checklist of the Reporting Summary provided by the editor. We also tried to elaborate on the composition of every subgroup labeled in the figures (particularly in the supplement), as suggested by the reviewer.

Reviewer #4 (Remarks to the Author):

We thank this reviewer for their contribution to the peer-reviewing process in collaboration to one of the previous reviewers. We hope their concerns have been successfully addressed in this rebuttal.

REFERENCES

- Alberich-Jordà, M. *et al.* (2012) 'C/EBP γ deregulation results in differentiation arrest in acute myeloid leukemia.', *The Journal of clinical investigation*, 122(12), pp. 4490–504. Available at: <https://doi.org/10.1172/JCI65102>.
- Andersson, R. *et al.* (2014) 'An atlas of active enhancers across human cell types and tissues', *Nature*, 507(7493), pp. 455–461. Available at: <https://doi.org/10.1038/nature12787>.
- Arber, D.A. *et al.* (2016) 'The 2016 revision to the World Health Organization classification of myeloid neoplasms and acute leukemia', *Blood*. American Society of Hematology, pp. 2391–2405. Available at: <https://doi.org/10.1182/blood-2016-03-643544>.
- Avellino, R. *et al.* (2016) 'An autonomous CEBPA enhancer specific for myeloid-lineage priming and neutrophilic differentiation', *Blood*, 127(24), pp. 2991–3003. Available at: <https://doi.org/10.1182/blood-2016-01-695759>.
- Bock, C. *et al.* (2012) 'DNA Methylation Dynamics during In Vivo Differentiation of Blood and Skin Stem Cells', *Molecular Cell*, 47(4), pp. 633–647. Available at: <https://doi.org/10.1016/j.molcel.2012.06.019>.
- Bormann, F. *et al.* (2018) 'Cell-of-Origin DNA Methylation Signatures Are Maintained during Colorectal Carcinogenesis', *Cell Reports*, 23(11), pp. 3407–3418. Available at: <https://doi.org/10.1016/j.celrep.2018.05.045>.
- Borssén, M. *et al.* (2013) 'Promoter DNA methylation pattern identifies prognostic subgroups in childhood T-cell acute lymphoblastic leukemia', *PLoS one*, 8(6). Available at: <https://doi.org/10.1371/JOURNAL.PONE.0065373>.
- Buck-Koehn, B.A. and Defossez, P.A. (2013) 'On how mammalian transcription factors recognize methylated DNA', *Epigenetics*, 8(2), pp. 131–137. Available at: <https://doi.org/10.4161/epi.23632>.
- Close, V. *et al.* (2019) 'FBXW7 mutations reduce binding of NOTCH1, leading to cleaved NOTCH1 accumulation and target gene activation in CLL', *Blood*, 133(8), pp. 830–839. Available at: <https://doi.org/10.1182/BLOOD-2018-09-874529>.
- Coustan-Smith, E. *et al.* (2009) 'Early T-cell precursor leukaemia: a subtype of very high-risk acute lymphoblastic leukaemia', *The Lancet Oncology*, 10(2), pp. 147–156. Available at: [https://doi.org/10.1016/S1470-2045\(08\)70314-0](https://doi.org/10.1016/S1470-2045(08)70314-0).
- Englesberg, E. *et al.* (1965) 'Positive control of enzyme synthesis by gene C in the L-arabinose system.', *Journal of Bacteriology*, 90(4), pp. 946–957. Available at: <https://doi.org/10.1128/JB.90.4.946-957.1965>.
- Englesberg, E. and Wilcox, G. (1974) 'Regulation: positive control.', *Annual review of genetics*, pp. 219–242. Available at: <https://doi.org/10.1146/annurev.ge.08.120174.001251>.
- Ernst, J. *et al.* (2011) 'Mapping and analysis of chromatin state dynamics in nine human cell types', *Nature*, 473(7345), pp. 43–49. Available at: <https://doi.org/10.1038/nature09906>.
- Fagerland, M.W. (2012) 'T-tests, non-parametric tests, and large studies: a paradox of statistical practice?', *BMC Medical Research Methodology*, 12(1), pp. 1–7. Available at: <https://doi.org/10.1186/1471-2288-12-78>.
- Farlik, M. *et al.* (2016) 'DNA Methylation Dynamics of Human Hematopoietic Stem Cell Differentiation', *Cell Stem Cell*, 19(6), pp. 808–822. Available at: <https://doi.org/10.1016/j.stem.2016.10.019>.
- Figuroa, M.E. *et al.* (2009) 'Genome-wide epigenetic analysis delineates a biologically distinct immature acute leukemia with myeloid/T-lymphoid features', *Blood*, 113(12), pp. 2795–2804. Available at: <https://doi.org/10.1182/blood-2008-08-172387>.
- Filarsky, K. *et al.* (2016) 'Krüppel-like factor 4 (KLF4) inactivation in chronic lymphocytic leukemia correlates with promoter DNA-methylation and can be reversed by inhibition of NOTCH signaling', *Haematologica*. Ferrata Storti Foundation, pp. e249–e253. Available at: <https://doi.org/10.3324/haematol.2015.138172>.
- Fishilevich, S. *et al.* (2017) 'GeneHancer: Genome-wide integration of enhancers and target genes in GeneCards', *Database*, 2017. Available at: <https://doi.org/10.1093/database/bax028>.
- Fulco, C.P. *et al.* (2019) 'Activity-by-contact model of enhancer–promoter regulation from thousands of CRISPR perturbations', *Nature Genetics*. Nature Publishing Group, pp. 1664–1669. Available at: <https://doi.org/10.1038/s41588-019-0538-0>.

Garcia-Manero, G. *et al.* (2002) 'DNA methylation of multiple promoter-associated CpG islands in adult acute lymphocytic leukemia.', *Clinical cancer research : an official journal of the American Association for Cancer Research*, 8(7), pp. 2217–24. Available at: <https://doi.org/10.1016/J.XGEN.2022.100112>.

Gasparini, M. *et al.* (2019) 'A Genome-wide Framework for Mapping Gene Regulation via Cellular Genetic Screens', *Cell*, 176(1–2), pp. 377–390.e19. Available at: <https://doi.org/10.1016/j.cell.2018.11.029>.

Gaston, K. and Fried, M. (1995) 'CpG methylation has differential effects on the binding of YY1 and ETS proteins to the bi-directional promoter of the Surf-1 and surf-2 genes', *Nucleic Acids Research*, 23(6), pp. 901–909. Available at: <https://doi.org/10.1093/nar/23.6.901>.

Gebhard, C. *et al.* (2019) 'Profiling of aberrant DNA methylation in acute myeloid leukemia reveals subclasses of CG-rich regions with epigenetic or genetic association', *Leukemia*, 33(1), pp. 26–36. Available at: <https://doi.org/10.1038/s41375-018-0165-2>.

Di Giammartino, D.C. *et al.* (2019) 'KLF4 is involved in the organization and regulation of pluripotency-associated three-dimensional enhancer networks', *Nature Cell Biology*, 21(10), pp. 1179–1190. Available at: <https://doi.org/10.1038/s41556-019-0390-6>.

Glass, J.L. *et al.* (2017) 'Epigenetic identity in AML depends on disruption of nonpromoter regulatory elements and is affected by antagonistic effects of mutations in epigenetic modifiers', *Cancer Discovery*, 7(8), pp. 868–883. Available at: <https://doi.org/10.1158/2159-8290.CD-16-1032>.

Gómez-de-Mariscal, E. *et al.* (2021) 'Use of the p-values as a size-dependent function to address practical differences when analyzing large datasets', *Scientific Reports*, 11(1), pp. 1–13. Available at: <https://doi.org/10.1038/s41598-021-00199-5>.

Govaerts, I. *et al.* (2021) 'PSEN1-selective gamma-secretase inhibition in combination with kinase or XPO-1 inhibitors effectively targets T cell acute lymphoblastic leukemia', *Journal of Hematology and Oncology*, 14(1), pp. 1–14. Available at: <https://doi.org/10.1186/s13045-021-01114-1>.

Halsey, L.G. (2019) 'The reign of the p-value is over: What alternative analyses could we employ to fill the power vacuum?', *Biology Letters*. The Royal Society. Available at: <https://doi.org/10.1098/rsbl.2019.0174>.

Harrington, M.A. *et al.* (1988) 'Cytosine methylation does not affect binding of transcription factor Sp1', *Proceedings of the National Academy of Sciences of the United States of America*, 85(7), pp. 2066–2070. Available at: <https://doi.org/10.1073/pnas.85.7.2066>.

Hasemann, M.S. *et al.* (2014) 'C/EBPα Is Required for Long-Term Self-Renewal and Lineage Priming of Hematopoietic Stem Cells and for the Maintenance of Epigenetic Configurations in Multipotent Progenitors', *PLoS Genetics*, 10(1). Available at: <https://doi.org/10.1371/journal.pgen.1004079>.

Hock, H. *et al.* (2004) 'Tel/Etv6 is an essential and selective regulator of adult hematopoietic stem cell survival', *Genes and Development*, 18(19), pp. 2336–2341. Available at: <https://doi.org/10.1101/gad.1239604>.

Hu, M. *et al.* (1997) 'Multilineage gene expression precedes commitment in the hemopoietic system', *Genes and Development*, 11(6), pp. 774–785. Available at: <https://doi.org/10.1101/gad.11.6.774>.

Hughes, L.A.E. *et al.* (2013) 'The CpG island methylator phenotype: What's in a name?', *Cancer Research*. American Association for Cancer Research, pp. 5858–5868. Available at: <https://doi.org/10.1158/0008-5472.CAN-12-4306>.

Ichim, C. V. *et al.* (2018) 'The orphan nuclear receptor EAR-2 (NR2F6) inhibits hematopoietic cell differentiation and induces myeloid dysplasia in vivo', *Biomarker Research*, 6(1), pp. 1–14. Available at: <https://doi.org/10.1186/s40364-018-0149-4>.

Jacob, F. and Monod, J. (1961) 'Genetic regulatory mechanisms in the synthesis of proteins', *Journal of Molecular Biology*, pp. 318–356. Available at: [https://doi.org/10.1016/S0022-2836\(61\)80072-7](https://doi.org/10.1016/S0022-2836(61)80072-7).

Ji, H. *et al.* (2010) 'Comprehensive methylome map of lineage commitment from haematopoietic progenitors', *Nature*, 467(7313), pp. 338–342. Available at: <https://doi.org/10.1038/nature09367>.

Kelly, A.D. *et al.* (2017) 'A CpG island methylator phenotype in acute myeloid leukemia independent of IDH mutations and associated with a favorable outcome', *Leukemia*, 31(10), pp. 2011–2019. Available at:

<https://doi.org/10.1038/leu.2017.12>.

Kim, M. and Costello, J. (2017) 'DNA methylation: An epigenetic mark of cellular memory', *Experimental and Molecular Medicine*. Nature Publishing Group, pp. e322–e322. Available at: <https://doi.org/10.1038/emm.2017.10>.

Kulesa, H., Frampton, J. and Graf, T. (1995) 'GATA-1 reprograms avian myelomonocytic cell lines into eosinophils, thromboblats, and erythroblats', *Genes and Development*, 9(10), pp. 1250–1262. Available at: <https://doi.org/10.1101/gad.9.10.1250>.

Laiosa, C. V. *et al.* (2006) 'Reprogramming of Committed T Cell Progenitors to Macrophages and Dendritic Cells by C/EBP α and PU.1 Transcription Factors', *Immunity*, 25(5), pp. 731–744. Available at: <https://doi.org/10.1016/j.immuni.2006.09.011>.

Longabaugh, W.J.R. *et al.* (2017) 'Bcl11b and combinatorial resolution of cell fate in the T-cell gene regulatory network', *Proceedings of the National Academy of Sciences of the United States of America*, 114(23), pp. 5800–5807. Available at: <https://doi.org/10.1073/pnas.1610617114>.

López-Guerra, M. *et al.* (2015) 'The γ -secretase inhibitor PF-03084014 combined with fludarabine antagonizes migration, invasion and angiogenesis in NOTCH1-mutated CLL cells', *Leukemia*, 29(1), pp. 96–106. Available at: <https://doi.org/10.1038/leu.2014.143>.

Melki, J.R., Vincent, P.C. and Clark, S.J. (1999) 'Concurrent DNA hypermethylation of multiple genes in acute myeloid leukemia', *Cancer Research*, 59(15), pp. 3730–3740. Available at: <https://aacrjournals.org/cancerres/article/59/15/3730/505321/Concurrent-DNA-Hypermethylation-of-Multiple-Genes> (Accessed: 23 March 2022).

Nerlov, C. and Graf, T. (1998) 'PU.1 induces myeloid lineage commitment in multipotent hematopoietic progenitors', *Genes & development*, 12(15), pp. 2403–2412. Available at: <https://doi.org/10.1101/GAD.12.15.2403>.

Newman, A.M. *et al.* (2019) 'Determining cell type abundance and expression from bulk tissues with digital cytometry', *Nature Biotechnology*, 37(7), pp. 773–782. Available at: <https://doi.org/10.1038/s41587-019-0114-2>.

O'Neil, J. *et al.* (2007) 'FBW7 mutations in leukemic cells mediate NOTCH pathway activation and resistance to γ -secretase inhibitors', *Journal of Experimental Medicine*, 204(8), pp. 1813–1824. Available at: <https://doi.org/10.1084/jem.20070876>.

Orkin, S.H. and Zon, L.I. (2008) 'Hematopoiesis: An Evolving Paradigm for Stem Cell Biology', *Cell*. Elsevier, pp. 631–644. Available at: <https://doi.org/10.1016/j.cell.2008.01.025>.

Pliner, H.A. *et al.* (2018) 'Cicero Predicts cis-Regulatory DNA Interactions from Single-Cell Chromatin Accessibility Data', *Molecular Cell*, 71(5), pp. 858–871.e8. Available at: <https://doi.org/10.1016/j.molcel.2018.06.044>.

Rhodes, J. *et al.* (2005) 'Interplay of pu.1 and gata1 determines myelo-erythroid progenitor cell fate in zebrafish', *Developmental cell*, 8(1), pp. 97–108. Available at: <https://doi.org/10.1016/j.DEVCEL.2004.11.014>.

Rishi, V. *et al.* (2010) 'CpG methylation of half-CRE sequences creates C/EBP α binding sites that activate some tissue-specific genes', *Proceedings of the National Academy of Sciences of the United States of America*, 107(47), pp. 20311–20316. Available at: https://doi.org/10.1073/PNAS.1008688107/SUPPL_FILE/APPENDIX.PDF.

Roman-Gomez, J. *et al.* (2005) 'Lack of CpG island methylator phenotype defines a clinical subtype of T-cell acute lymphoblastic leukemia associated with good prognosis', *Journal of clinical oncology : official journal of the American Society of Clinical Oncology*, 23(28), pp. 7043–7049. Available at: <https://doi.org/10.1200/JCO.2005.01.4944>.

Sanyal, A. *et al.* (2012) 'The long-range interaction landscape of gene promoters', *Nature*, 489(7414), pp. 109–113. Available at: <https://doi.org/10.1038/nature11279>.

Sawilowsky, S.S. (2009) 'Very large and huge effect sizes', *Journal of Modern Applied Statistical Methods*, 8(2), pp. 597–599. Available at: <https://doi.org/10.22237/jmasm/1257035100>.

Schep, A.N. *et al.* (2017) 'chromVAR: inferring transcription-factor-associated accessibility from single-cell epigenomic data', *Nature Methods* [Preprint]. Available at: <https://doi.org/10.1038/nmeth.4401>.

Shan, Q. *et al.* (2021) 'Tcf1 and Lef1 provide constant supervision to mature CD8+ T cell identity and function by organizing genomic architecture', *Nature Communications*, 12(1), pp. 1–20. Available at: <https://doi.org/10.1038/s41467-021-26159->

1.

Somerville, T.D.D. *et al.* (2018) 'Derepression of the Iroquois Homeodomain Transcription Factor Gene IRX3 Confers Differentiation Block in Acute Leukemia', *Cell Reports*, 22(3), p. 638. Available at: <https://doi.org/10.1016/J.CELREP.2017.12.063>.

Soto, L.F. *et al.* (2021) 'Compendium of human transcription factor effector domains', *Molecular Cell* [Preprint]. Available at: <https://doi.org/10.1016/j.molcel.2021.11.007>.

Souroullas, G.P. *et al.* (2009) 'Adult Hematopoietic Stem and Progenitor Cells Require Either Lyl1 or Scl for Survival', *Cell Stem Cell*, 4(2), pp. 180–186. Available at: <https://doi.org/10.1016/j.stem.2009.01.001>.

Stresemann, C. and Lyko, F. (2008) 'Modes of action of the DNA methyltransferase inhibitors azacytidine and decitabine', *International Journal of Cancer*. John Wiley & Sons, Ltd, pp. 8–13. Available at: <https://doi.org/10.1002/ijc.23607>.

Thompson, B.J. *et al.* (2007) 'The SCFFBW7 ubiquitin ligase complex as a tumor suppressor in T cell leukemia', *The Journal of Experimental Medicine*, 204(8), p. 1825. Available at: <https://doi.org/10.1084/JEM.20070872>.

Touzart, A. *et al.* (2021) 'Epigenetic analysis of patients with T-ALL identifies poor outcomes and a hypomethylating agent-responsive subgroup', *Science Translational Medicine*, 13(595). Available at: <https://doi.org/10.1126/scitranslmed.abc4834>.

Toyota, M. *et al.* (1999) 'CpG island methylator phenotype in colorectal cancer', *Proceedings of the National Academy of Sciences of the United States of America*, 96(15), pp. 8681–8686. Available at: <https://doi.org/10.1073/pnas.96.15.8681>.

Toyota, M. *et al.* (2001) 'Methylation profiling in acute myeloid leukemia', *Blood*, 97(9), pp. 2823–2829. Available at: <https://doi.org/10.1182/blood.V97.9.2823>.

Wei, Z. *et al.* (2013) 'Klf4 organizes long-range chromosomal interactions with the OCT4 locus in reprogramming and pluripotency', *Cell Stem Cell*, 13(1), pp. 36–47. Available at: <https://doi.org/10.1016/j.stem.2013.05.010>.

Weng, A.P. *et al.* (2004) 'Activating mutations of NOTCH1 in human T cell acute lymphoblastic leukemia', *Science*, 306(5694), pp. 269–271. Available at: <https://doi.org/10.1126/science.1102160>.

Wouters, B.J. *et al.* (2007) 'Distinct gene expression profiles of acute myeloid/T-lymphoid leukemia with silenced CEBPA and mutations in NOTCH1', *Blood*, 110(10), pp. 3706–3714. Available at: <https://doi.org/10.1182/blood-2007-02-073486>.

Xiang, P. *et al.* (2010) 'Identification of E74-like factor 1 (ELF1) as a transcriptional regulator of the Hox cofactor MEIS1', *Experimental Hematology*, 38(9), pp. 798–808.e2. Available at: <https://doi.org/10.1016/j.exphem.2010.06.006>.

Yan, F. *et al.* (2020) 'From reads to insight: A hitchhiker's guide to ATAC-seq data analysis', *Genome Biology*, 21(1), pp. 1–16. Available at: <https://doi.org/10.1186/s13059-020-1929-3>.

Yin, Y. *et al.* (2017) 'Impact of cytosine methylation on DNA binding specificities of human transcription factors', *Science*, 356(6337). Available at: <https://doi.org/10.1126/science.aaj2239>.

Zeitlinger, J. (2020) 'Seven myths of how transcription factors read the cis-regulatory code', *Current Opinion in Systems Biology*. Elsevier, pp. 22–31. Available at: <https://doi.org/10.1016/j.coisb.2020.08.002>.

Zhu, H., Wang, G. and Qian, J. (2016) 'Transcription factors as readers and effectors of DNA methylation', *Nature Reviews Genetics*, 17(9), pp. 551–565. Available at: <https://doi.org/10.1038/nrg.2016.83>.

Zhu, T. *et al.* (2022) 'A pan-tissue DNA methylation atlas enables in silico decomposition of human tissue methylomes at cell-type resolution', *Nature Methods*, 19(3), pp. 296–306. Available at: <https://doi.org/10.1038/s41592-022-01412-7>.

Reviewers' Comments:

Reviewer #1:

Remarks to the Author:

The authors have adressed my concerns in a satisfactory way.

Reviewer #2:

Remarks to the Author:

The authors have addressed all my concerns. Fantastic work!